# PI-Light: Physics-Inspired Diffusion for Full-Image Relighting

**Zhexin Liang**[1], **Zhaoxi Chen**[1], **Yongwei Chen**[1],
**Tianyi Wei**[1], **Tengfei Wang**[2], **and Xingang Pan**[1]
[1]S-Lab, Nanyang Technological University [2]Tencent

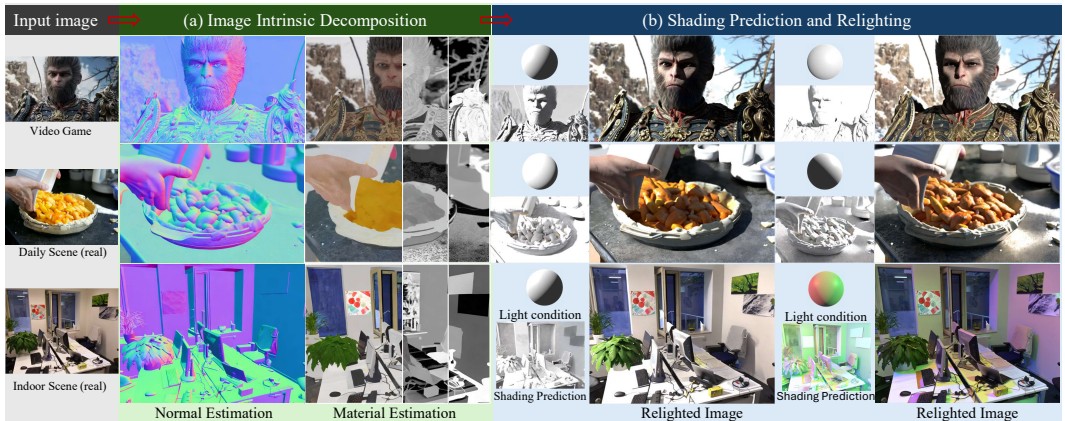

Figure 1: $\pi$-**Light enables full-image relighting through physics-inspired diffusion models.** It achieves favorable results in both a) inverse rendering and b) neural forward rendering stages, enabling precise control over lighting.

## Abstract

Full-image relighting remains a challenging problem due to the difficulty of collecting large-scale structured paired data, the difficulty of maintaining physical plausibility, and the limited generalizability imposed by data-driven priors. Existing attempts to bridge the synthetic-to-real gap for full-scene relighting remain suboptimal. To tackle these challenges, we introduce **P**hysics-**I**nspired diffusion for full-image re**Light**ing ($\pi$-Light, or PI-Light), a two-stage framework that leverages physics-inspired diffusion models. Our design incorporates (i) batch-aware attention, which improves the consistency of intrinsic predictions across a collection of images, (ii) a physics-guided neural rendering module that enforces physically plausible light transport, (iii) physics-inspired losses that regularize training dynamics toward a physically meaningful landscape, thereby enhancing generalizability to real-world image editing, and (iv) a carefully curated dataset of diverse objects and scenes captured under controlled lighting conditions. Together, these components enable efficient finetuning of pretrained diffusion models while also providing a solid benchmark for downstream evaluation. Experiments demonstrate that $\pi$-Light synthesizes specular highlights and diffuse reflections across a wide variety of materials, achieving superior generalization to real-world scenes compared with prior approaches. Code will be available at https://github.com/ZhexinLiang/PI-Light.

## 1 Introduction

Recent advances in computer graphics and vision have highlighted image relighting as a critical task for film production, augmented reality, and digital content creation. Existing works can be grouped into object-centric approaches (Zeng et al., 2024a; Mei et al., 2023; Zhang et al., 2025; Ponglertnapakorn et al., 2023; He et al., 2024; Futschik et al., 2023), which often ignore or replace

the background, and scene-level methods (Choi et al., 2024; Kocsis et al., 2024a), which attempt full-image relighting but remain largely data-driven and struggle to generalize beyond the training distribution without extensive diverse data, and cannot maintain self-luminous objects' lighting. While state-of-the-art foreground relighting methods (Zhang et al., 2025) have shown impressive results, they still suffer from albedo inconsistency and imprecise lighting control. For example, as shown in Fig. 2, carpet colors are altered ($1^{st}$ row) and the intended leftward light appears to come from the left rear in the absence of an internal light source ($2^{nd}$ row). These limitations highlight the lack of a decomposed scene interpretation and the inherent limitations of data-driven priors, which are essential for controllable and physically plausible full-image relighting.

While prior methods have advanced the field, their limitations highlight three major challenges for full-image relighting. First, the main challenge lies in the difficulty of acquiring datasets with diverse and complex scene illuminations: obtaining high-quality images under diverse and controlled lighting conditions across full scenes is both resource-intensive and technically challenging. In particular, it is infeasible to obtain real-world datasets that capture the same scene under multiple lighting conditions. Second, ensuring physical plausibility remains challenging, as purely data-driven approaches often

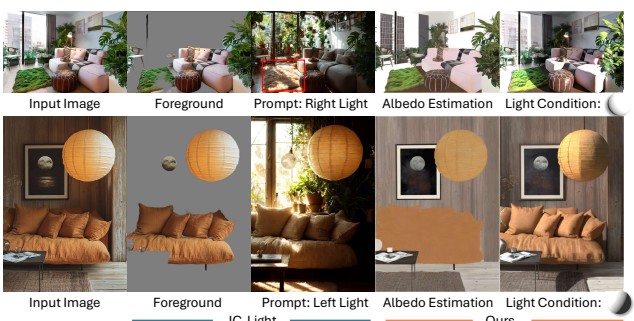

Figure 2: **Comparisons between IC-Light (Zhang et al., 2025) and Ours.** The state-of-the-art foreground relighting method still suffers from inconsistencies in maintaining albedo uniformity and lacks physically plausible control over lighting direction.

generate results that violate basic light transport principles. Finally, limited data-driven priors restrict generalizability, leading to poor robustness when facing more unseen materials, lighting conditions, or realistic scene compositions.

To alleviate these challenges, we propose $\pi$-Light, a novel diffusion-based neural relighting pipeline designed to tackle the full-image relighting problem. Similar to the previous PBR approach (Zeng et al., 2024b; Liang et al., 2025), our approach also adopts a two-stage framework of inverse neural rendering followed by neural forward rendering, but differs from prior works in the following key aspects. First, inspired by Wonder3D (Long et al., 2024), during both inverse neural rendering and neural forward rendering stages, we extend standard self-attention layers to be global-aware, enabling communication across batches. This design improves both efficiency and consistency among predicted intrinsics. Second, we design a physics-inspired neural forward rendering module, where the physics-based loss serves as an efficient learning mechanism. Note that, our physics-inspired loss function is non-trivial. It regularizes the training dynamics into a physically plausible landscape, enabling easier convergence and allowing the model to learn more correct light transport with less data and computation. As a result, our method achieves competitive performance and generalizes well to real-world scenes, even with fewer training samples than previous works (Zeng et al., 2024b; Liang et al., 2025) and without access to fully realistic datasets. We further propose a simple yet effective lighting representation that uses only the front hemisphere of the environment map. This design avoids interference from self-luminous objects and built-in scene lighting, enabling effective control of light direction and intensity while preserving background consistency.

To address the data scarcity that hinders full-image relighting research, we constructed a new dataset featuring diverse objects and scenes, all captured under controlled lighting conditions. This dataset not only supports supervised learning but also enables comprehensive downstream benchmarking. By training on this dataset, combined with the image prior from pretraining, $\pi$-Light demonstrates strong generalization across diverse objects and scenes, as extensively validated in our experiments.

Our contributions can be summarized as follows:

- We propose $\pi$-Light, a diffusion-based neural relighting framework that achieves physically-aware full-image relighting.

- We impose physics-inspired light transport priors onto the neural forward rendering module as a regulation to help the model converge toward physical principles and enhance generalizability.

- We contribute a new, high-quality dataset of diverse objects and scenes rendered under controlled lighting conditions, providing a valuable resource for advancing full-image relighting research.

- Extensive experiments demonstrate the superiority of our method over previous models.

## 2 RELATED WORK

**Neural Rendering.** PBR simulation has recently emerged as a research focus, encompassing both inverse rendering and forward rendering processes. Many recent works (Zhu et al., 2022; Zhang et al., 2024; Zeng et al., 2024a; Kocsis et al., 2024b; Li et al., 2020) have concentrated on the inverse rendering process, also referred to as image intrinsic decomposition. However, due to the difficulty of acquiring large-scale training data, these methods have limited applicability and are primarily designed for specific scenarios, such as indoor scenes (Zhu et al., 2022; Kocsis et al., 2024b; Li et al., 2020) or object-level inverse rendering (Zhang et al., 2024; Zeng et al., 2024a; Jin et al., 2024). MAIR (Choi et al., 2023) improves inverse rendering with multi-view attention and spatially-varying lighting estimation but is sensitive to viewpoint inconsistencies.

Other approaches focus on the inverse rendering of illumination maps (Enyo & Nishino, 2024; Lyu et al., 2023), aiming to infer scene lighting properties directly from images. Additionally, some methods (Bae & Davison, 2024; Bae et al., 2021; Fu et al., 2024) specialize in geometry estimation, particularly in normal and depth estimation. Furthermore, unsupervised approaches and those leveraging large-scale model priors (Pan et al., 2020; Papantoniou et al., 2023; Chen et al., 2024) introduce various constraints to recover multiple intrinsic components.

RGB↔X (Zeng et al., 2024b) successfully employs a diffusion model to simulate both the inverse and forward rendering processes. However, its forward rendering relies on irradiance as the input lighting condition, which makes precise lighting control challenging. Additionally, its application is limited to indoor scenes. More recent work (Liang et al., 2025) leverages a video diffusion model to perform both inverse and forward rendering of videos. We further discuss the differences between our method and these two approaches in Appendix A.11.1.

**2D Image Relighting.** In addition to advances in 3D relighting, such as Poirier-Ginter et al. (2024); Zhao et al. (2024), 2D relighting has also begun to attract increasing attention in recent years. Recent work (Zhang et al., 2025) has achieved relighting effects for foreground objects and can generate different scenes based on prompts. However, as shown in Fig. 2, its data-driven approach without physical representations still fails to preserve the albedo and background, lacking precise controls over lighting direction.

Many previous studies focus on portrait relighting (Papantoniou et al., 2023; Pandey et al., 2021; Tajima et al., 2021; Futschik et al., 2023; Ponglertnapakorn et al., 2023; Mei et al., 2023) or object relighting (Bashkirova et al., 2023), achieving practical performance in foreground relighting. ScribbleLight (Choi et al., 2024) is a model that enables indoor scene relighting using scribble-based controls. However, applying scribbles to control lighting directly on images still lacks precision and it's application is limited to indoor scenes. While LightIt (Kocsis et al., 2024a) and OutCast (Griffiths et al., 2022) enable outdoor full-image relighting, LightIt requires normals as input, and both methods rely on directional lighting models that generalize poorly to indoor scenes.

## 3 PRELIMINARIES

**Physically-Based Rendering (PBR)** is a rendering approach that simulates interactions of light with materials in a physically accurate manner. The **Principled BRDF** (based on the Disney Principled Shader (Burley & Studios, 2012)) is a user-friendly BRDF model, which is based on microfacet BRDF models, including the Lambertian reflection model for diffuse surfaces and the Cook-Torrance microfacet model for specular reflections. Here, we only introduce its core formula to

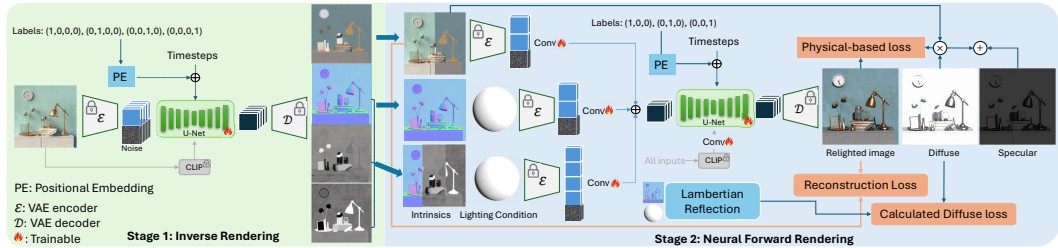

Figure 3: **Overview of $\pi$-Light. Stage 1: Inverse Neural Rendering.** Given a 2D image as input, this stage repurposes a pretrained image diffusion model to simultaneously predict four intrinsic components: albedo, normal, roughness, and metallic. **Stage 2: Neural Forward Rendering.** Given the input image, the physical intrinsics from Stage 1, and a target lighting condition, this stage tames the image diffusion model to generate the relit image along with its diffuse and specular shading, guided by a physics-inspired light transport prior.

compute the diffuse map:

$$D(p) = \int_\Omega L(\omega) \max(0, N(p) \cdot \omega) d\omega = \int_\Omega L(\omega) \max(0, N(p) \cdot \omega) sin\theta \cdot d\theta d\phi, \qquad (1)$$

where $p$ is a pixel location in image, $\Omega$ is the hemisphere above the surface, $L(\omega)$ is the environment map lighting in the incoming light direction $\omega$, $N(p)$ is the surface normal at $p$.

**Latent Diffusion Model (LDM).** LDM (Rombach et al., 2022) is proposed to run the diffusion process in latent space instead of pixel space, saving computational costs. LDM introduces an autoencoder model for image reconstruction. The encoder $\mathcal{E}$ encodes the input image in a 2D latent space $\mathbf{z}$. The decoder $\mathcal{D}$ then reconstructs the image from $\mathbf{z}$. Thus, the overall LDM framework consists of a denoising network (U-Net) and an autoencoder. We calculate the overall loss using V-prediction (Salimans & Ho, 2022):

$$L_{\text{V-pred}} = \mathbb{E}_{z_0, \epsilon, t} \left[ \| \hat{v}_\theta(z_t, t) - v_t \|_2^2 \right], \qquad (2)$$

where, $$z_t = \sqrt{\bar{\alpha}_t} z_0 + \sqrt{1 - \bar{\alpha}_t} \epsilon, \quad v_t = \sqrt{\bar{\alpha}_t} \epsilon - \sqrt{1 - \bar{\alpha}_t} z_0, \qquad (3)$$

$\hat{v}_\theta(x_t, t)$ is the model's predicted V-prediction, and the expectation is taken over data samples $z_0$, Gaussian noise $\epsilon$, and timestep $t$.

## 4 METHODOLOGY

In this section, we present our dataset construction pipeline along with our two-stage method. First, in Sec. 4.1, we introduce our data construction pipeline, which encompasses both object- and scene-level data. The dataset consists of RGB images paired with their corresponding ground-truth intrinsic properties, as well as light conditions, rendered using Blender (Foundation, 2024). Unlike previous methods (Kocsis et al., 2024b; Zeng et al., 2024b) that represent illumination using irradiance, we employ a rendered gray ball to model lighting, allowing for more precise and intuitive user control. Next, in Sec. 4.2, leveraging the constructed dataset, we describe our inverse neural rendering pipeline, which decomposes an input RGB image into multiple intrinsic components. Finally, in Sec. 4.3, utilizing the decomposed intrinsics from the previous stage, we finetune Stable Diffusion (Rombach et al., 2022) to relight the image based on user-provided lighting conditions. An overview of our method is illustrated in Fig. 3.

### 4.1 DATASET CONSTRUCTION

To train a model capable of accurately decomposing an input RGB image, we need to collect objects and scenes that contain rich material information. At the object level, we filter objects from Objaverse (Deitke et al., 2023b;a), selecting those with BRDF material properties. For scene data, while some existing datasets (Kocsis et al., 2024b; Zeng et al., 2024b) provide intrinsic components, they primarily focus on indoor scenes and lack object masks to exclude translucent or transparent objects,

which can hinder accurate material estimation. Moreover, these datasets do not render images under varying lighting conditions—a critical factor for our task. To address these limitations, we curate a scene dataset from an online repository (BlenderKit, 2024) and render intrinsic components along with diverse lighting conditions for training.

**Object Data.** For object-centric data, we randomly select over 10,000 objects from filtered Objaverse (Deitke et al., 2023b;a) that contain BRDF materials and render them under varying viewpoints and lighting conditions. Each object is rendered with 10 views and 10 lighting conditions, totaling 100 images per object. Lighting is sampled with a preference toward the upper hemisphere, and includes both point lights and HDRI maps randomly selected from over 700 environment maps from Poly Haven (Haven, 2025).

BlenderProc (Denninger et al., 2023) utilizes Blender's composition layer to generate intrinsic properties. However, we observed that certain albedo values fluctuate with the light source, and for transparent or semi-transparent materials, the composition layer often outputs entirely white or black values, which are inaccurate. To mitigate these issues, we extract all intrinsics directly from Blender's Principled BRDF nodes while generating a mask to exclude non-Principled regions.

Different from all previous methods (Zhang et al., 2025; Bashkirova et al., 2023; Zeng et al., 2024a) that either condition on global HDRI lighting information (Zeng et al., 2024a) or infer lighting from background to relight foreground (Zhang et al., 2025; Bashkirova et al., 2023), our method constructs the lighting condition using only the light sources directly facing the image: the front hemisphere of a gray ball rendered from the HDRI. This representation mitigates the influence of self-luminous objects and built-in light sources (e.g., light from windows) present in the original scene.

**Scene Data.** We curate 300 high-quality indoor and outdoor scenes in total from BlenderKit (BlenderKit, 2024), manually filtered to remove scenes with excessive special effects, such as fog and rain. Camera views are sampled via constrained random perturbations to maintain valid viewpoints. Lighting is augmented by adding a point light source behind the camera, ensuring diverse shadow and highlight patterns. To maintain consistency in normal rendering, we fix the Blender version per scene.

Details of the sampling strategies and rendering settings are provided in *Appendix* A.1.

## 4.2 STAGE 1: INVERSE NEURAL RENDERING

### 4.2.1 PIPELINE

As shown in Fig. 3 (Left), inspired by Wonder3D (Long et al., 2024) and GeoWizard (Fu et al., 2024), our first stage takes four batchwise concatenated input images $I_{in} \in \mathbb{R}^{C \times H \times W}$ as condition channelwisely concatenating with noise and simultaneously predicts the albedo $A \in \mathbb{R}^{C \times H \times W}$, normal $N \in \mathbb{R}^{C \times H \times W}$, roughness $R \in \mathbb{R}^{C \times H \times W}$, metallic $M \in \mathbb{R}^{C \times H \times W}$ for the given input. The CLIP image embedding of the condition inputs go into the cross-attention layers as well. The standard self-attention layers are extended to be global-aware, enabling communication across batches. This design improves both efficiency and consistency among predicted intrinsics.

Attention links across the four batches ensure structural consistency among the predicted intrinsic components while enabling the sharing of global information, thereby improving prediction accuracy. Our model uses four one-hot labels $L \in \mathbb{N}_+^{1 \times 4}$ to control which batch outputs which intrinsic component. As shown in Fig. 5, compared to the latest and most relevant 2D rendering work, RGB↔X (Zeng et al., 2024b), our model can generate all four intrinsic components simultaneously while maintaining structural consistency.

### 4.2.2 TRAINING OBJECTIVE

As mentioned in Sec. 4.1, some intrinsic components are either inaccurate or difficult to obtain during data collection, including albedo for translucent/transparent objects, normal for the sky, and intrinsic components of objects rendered without the Principled BRDF model. To address this issue, we generate a corresponding mask alongside each dataset sample to annotate these unreliable regions. Here we use this mask to calculate mask loss. It is important to note that the latent space and

RGB space are not strictly pixel-aligned, though we have not yet found a better alternative. Please refer to *Appendix* A.12.1 for further analysis.

Specifically, given the latent V-prediction output of any intrinsic component, denoted as $v_{\text{pred}} \in \mathbb{R}^{4 \times \frac{H}{8} \times \frac{W}{8}}$, the masked latent loss is applied directly in the V-prediction space. The corresponding mask $m$ is downsampled by a factor of 8, yielding $m_{\text{z}} \in \mathbb{R}^{\frac{H}{8} \times \frac{W}{8}}$, which is then element-wise multiplied with both the predicted and ground-truth latents. The loss is defined as follows:

$$L_{\text{stage1}} = MSE(v_{\text{pred}} \cdot m_{\text{z}}, v_{\text{target}} \cdot m_{\text{z}}), \tag{4}$$

where $v_{\text{target}} \in \mathbb{R}^{4 \times \frac{H}{8} \times \frac{W}{8}}$ denotes the V-prediction of the latent code obtained by adding ground truth noise to the ground truth intrinsic latent.

### 4.3 STAGE 2: NEURAL FORWARD RENDERING

#### 4.3.1 PHYSICS-INSPIRED PIPELINE

As shown in Fig. 3 (Right), our relighting pipeline is designed based on the Surface Reflection Model, formulated in equation 5 within the framework of Physically-Based Rendering (PBR):

$$I_{\text{rendered}} = A \odot D + S, \tag{5}$$

where $I_{\text{rendered}}$ represents the rendered image, and $A$, $D$, $S$ represent the albedo, diffuse and specular maps, separately.

Our model follows the Principled BRDF (Zeng et al., 2024a). Both the input and output of our relighting model consist of three components: as described in equation 8, our model separately outputs the diffuse and specular components, and the final rendered relighted image is obtained through a loss constraint which is illustrated in Sec. 4.3.2.

According to the Principled BRDF in Sec. 3, the diffuse component is modeled using Lambertian reflection, which depends solely on the lighting conditions and surface normal. In contrast, the specular component follows the Cook-Torrance microfacet model, which is influenced by a combination of lighting conditions, surface normal, roughness, and metallic properties.

To account for these dependencies, our relighting model takes three batchwise concatenated input conditions, denoted as:

$$I_{\text{in}}^{\text{stage2}} = [I_{\text{in1}}, I_{\text{in2}}, I_{\text{in3}}], \tag{6}$$

$$I_{\text{in1}} = (I_{\text{in}}, A), \ I_{\text{in2}} = (N, L, m), \ I_{\text{in3}} = (N, L, M, R, m), \tag{7}$$

where $I_{\text{in1}} \in \mathbb{R}^{2C \times H \times W}$ represents the concatenation of the input image and albedo, $I_{\text{in2}} \in \mathbb{R}^{3C \times H \times W}$ corresponds to the surface normal, lighting condition, and mask. and $I_{\text{in3}} \in \mathbb{R}^{5C \times H \times W}$ includes the surface normal, lighting condition, metallic, roughness and mask.

Each of these input conditions concatenating with noise maps to a corresponding batchwise concatenated output, denoted as:

$$I_{\text{out}}^{\text{stage2}} = [I_{\text{relit}}, D_{\text{pred}}, S_{\text{pred}}], \tag{8}$$

where $I_{\text{relit}}$ represents the final rendered relighted image, $D_{\text{pred}}$ corresponds to the diffuse component, and $S_{\text{pred}}$ denotes the specular component.

Thus, the input and output for each batch are:

$$I_{in1} = (I_{in}, A) \rightarrow I_{relit}, I_{in2} = (N, L, m) \rightarrow D_{pred}, I_{in3} = (N, L, M, R, m) \rightarrow S_{pred}$$

This formulation ensures that the second and third batches of the model focus on the strength and structure of lighting and shadow, while the first batch focuses more on the final rendered color, allowing the outputs of the second and third batches to more closely follow the properties provided in those batches. For instance, the third batch enables the specular component to directly learn the correspondence between the specular map and the metallic information provided in the third input channel. As a result, our method effectively reconstructs highlights under the given conditions, shown in Fig. 7. Furthermore, by incorporating our Physics-inspired loss, as shown in Sec. 4.3.2, we enforce physical consistency in the final rendered output, ensuring that it adheres to the given lighting conditions.

### 4.3.2 TRAINING OBJECTIVES

Since physical laws lose their meaning when computed in the latent space, our physics-based losses for the relighting model are applied entirely in the RGB space, i.e., it is computed after the VAE decoder reconstructs the output, as mentioned in Sec. 3.

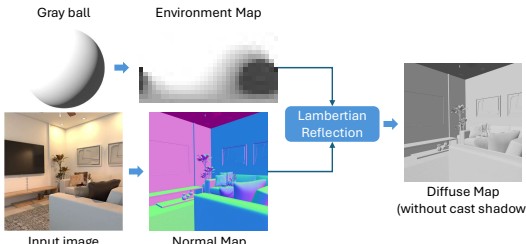

Figure 4: **Diffuse Shading Loss.** We incorporate a manually computed loss using the Lambertian model, without requiring ground truth annotations of diffuse shading.

**Diffuse Shading Loss.** Since obtaining ground-truth diffuse reflectance is challenging in certain cases and some scene lighting conditions may be inaccurate, we incorporate a manually-computed loss based on the normal map and environment map. This loss encourages the model to better capture the relationship between the normal and lighting conditions. The computation process is as follows:

As shown in Fig. 4, given a gray ball representing the light source $G$, we convert it into an environment map $E \in \mathbb{R}^{32 \times 16 \times 3}$. The normal map $N$ is then normalized to the range $[-1, 1]$.

Since the gray ball rendered in our scene already corresponds to a diffused version of the lighting representation, unfolding it directly yields the energy-conserving diffuse environment map $E_{\text{diff}}$, which is the discretized form of the diffuse map described in equation 1.

By lookup-based sampling using the normal map, we obtain the diffuse map that does not include cast shadows:

$$N = [\mathbf{n}_x, \mathbf{n}_y, \mathbf{n}_z]^T, \quad N \in \mathbb{R}^{3 \times H \times W}, \tag{9}$$

$$UV = [\frac{1}{\pi}tan^{-1}(\frac{\mathbf{n}_x}{\mathbf{n}_z}), \frac{2}{\pi}cos^{-1}(\mathbf{n}_y) - 1], \quad D_{\text{calculated}} = \text{grid\_sample}(E_{\text{diff}}, UV), \tag{10}$$

Finally, we compute the calculated diffuse loss between the calculated diffuse map $D_{\text{calculated}}$ and the predicted diffuse map $D_{\text{pred}}$ decoded from latent prediction:

$$L_{\text{DS}} = MSE(D_{\text{calculated}}, D_{\text{pred}}) \tag{11}$$

**Physical-based Shading Loss.** To ensure that the rendered results are shaded according to the given lighting conditions and adhere to physical principles, we define a physics-based loss as shown in Eq. 12, based on the formulation in Eq. 5.

$$L_{\text{PS}} = MSE(I_{\text{relit}}, A \odot D_{\text{pred}} + S_{\text{pred}}), \tag{12}$$

**Reconstruction Loss.** To ensure that the overall content of the image remains unchanged before and after relighting, we introduce a reconstruction loss. This loss is computed as a perceptual loss by extracting features from both images using the DINO feature extractor and measuring the difference between them. The specific formulation is given as follows:

$$L_{\text{rec}} = \|\phi(I_{\text{relit}}) - \phi(I_{\text{input}})\|_2^2, \tag{13}$$

where $\phi$ denotes the DINO feature extractor; $I_{\text{relit}}$ and $I_{\text{in}}$ denote the relighted and input images.

**Final loss function** for the training:

$$L_{\text{stage2}} = L_{\text{V-pred}} + \frac{1}{t}(\lambda_1 L_{\text{DS}} + \lambda_2 L_{\text{PS}} + \lambda_3 L_{\text{rec}}). \tag{14}$$

where $\lambda_1, \lambda_2, \lambda_3$ are three constants to weight the losses, $t$ is the timestep of the diffusion noise. We use $1/t$ to mitigate the errors introduced by directly inferring $z_0$ from the predicted $z_t$.

## 5 EXPERIMENTS

### 5.1 IMPLEMENTATION DETAILS

**Training.** We trained for $80k$ iterations in stage one and $90k$ iterations in stage two using eight 40GB A100 GPUs, with a batch size of 8 and a learning rate of $1e^{-5}$, optimized using Adam. The initial weights of the U-Net were derived from GeoWizard (Fu et al., 2024).

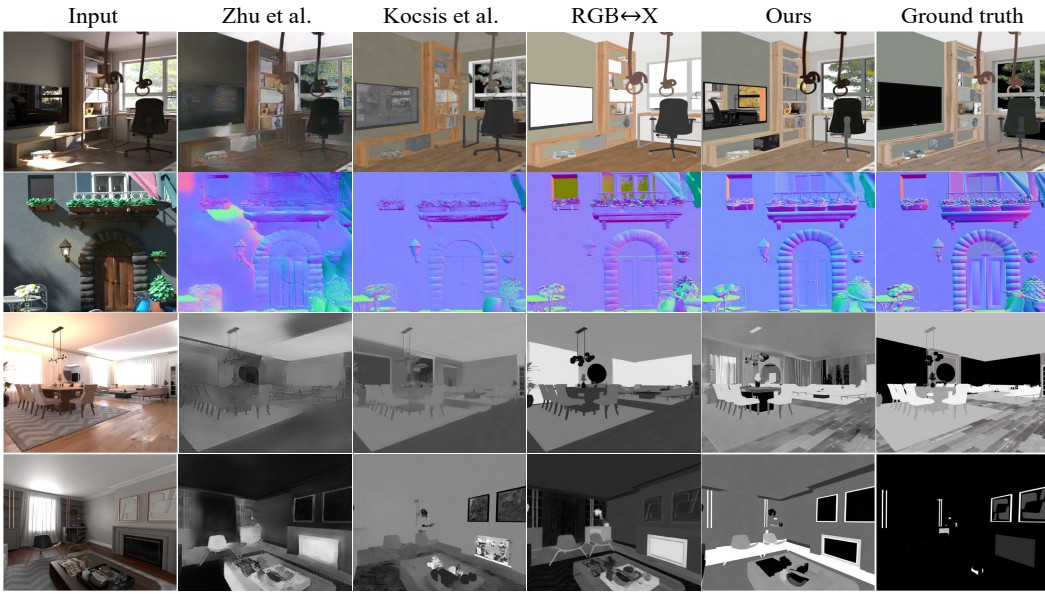

Input | Zhu et al. | Kocsis et al. | RGB↔X | Ours | Ground truth

*Part of the black regions in the ground truth represent masked areas, indicating that the shader used in these regions is not based on the Principled BRDF model and has therefore been masked. **(top to button: Albedo, Normal, Roughness, Metallic)**

Figure 5: Qualitative comparisons on our Scene200 test dataset. From top to bottom, the results correspond to albedo, normal, roughness, and metallic comparisons. Our method produces more detailed and accurate results, even in specular reflection regions.

| Dataset | Method | Albedo | | | Normal | | | Roughness | | | Metallic | | | Average | | |
|---|---|---|---|---|---|---|---|---|---|---|---|---|---|---|---|---|
| | | PSNR↑ | SSIM↑ | LPIPS↓ | PSNR↑ | SSIM↑ | LPIPS↓ | PSNR↑ | SSIM↑ | LPIPS↓ | PSNR↑ | SSIM↑ | LPIPS↓ | PSNR↑ | SSIM↑ | LPIPS↓ |
| Object50 | IntrinsicAnything (Chen et al., 2024) | 21.92 | 0.9157 | 0.0677 | - | - | - | - | - | - | - | - | - | - | - | - |
| | Kocsis et al.* (Kocsis et al., 2024b) | 20.62 | 0.8976 | 0.0796 | 24.30 | 0.9186 | 0.0665 | 17.66 | 0.8880 | 0.0859 | 14.02 | 0.8335 | 0.1250 | 19.15 | 0.8844 | 0.0893 |
| | Zhu et al.* (Zhu et al., 2022) | 19.45 | 0.8825 | 0.0801 | 21.87 | 0.8794 | 0.0849 | 15.89 | 0.8588 | 0.1013 | 12.49 | 0.8129 | 0.1505 | 17.43 | 0.8584 | 0.1042 |
| | RGB↔X (Zeng et al., 2024b) | 20.09 | 0.9128 | 0.0737 | 22.78 | 0.9039 | 0.0793 | 18.43 | **0.9061** | 0.0752 | **15.97** | **0.8404** | 0.1226 | 19.32 | 0.8908 | 0.0877 |
| | Ours | **22.09** | **0.9200** | **0.0562** | **24.97** | **0.9217** | **0.0531** | **21.09** | 0.8961 | **0.0736** | 13.96 | 0.8357 | **0.1216** | **20.53** | **0.8934** | **0.0761** |
| Scene200 | IntrinsicAnything (Chen et al., 2024) | 13.51 | 0.6280 | 0.3399 | - | - | - | - | - | - | - | - | - | - | - | - |
| | Kocsis et al.* (Kocsis et al., 2024b) | **14.69** | 0.6767 | 0.2952 | **18.17** | **0.7533** | 0.2710 | 10.44 | 0.6065 | 0.3381 | 5.55 | 0.2410 | **0.5856** | 12.21 | **0.5694** | 0.3778 |
| | Zhu et al.* (Zhu et al., 2022) | 13.87 | 0.6459 | 0.2565 | 16.24 | 0.6654 | **0.2696** | 9.64 | 0.5380 | 0.3229 | 5.40 | 0.2175 | 0.5972 | 11.29 | 0.4629 | **0.3616** |
| | RGB↔X (Zeng et al., 2024b) | 14.07 | 0.6955 | 0.2441 | 14.59 | 0.6824 | 0.3278 | 11.14 | 0.6187 | 0.3086 | **7.20** | **0.2217** | 0.6764 | 11.75 | 0.5546 | 0.3892 |
| | Ours | 14.46 | **0.7025** | **0.2248** | 16.26 | 0.7208 | 0.2768 | **14.05** | **0.6348** | **0.2980** | 7.02 | 0.2001 | 0.6966 | **12.95** | 0.5646 | 0.3741 |

* indicates that the resolution differs from the original resolution 768 × 768 of our test dataset. Kocsis et al.(Kocsis et al., 2024b) resize images to 480 × 640, while Zhu et al.(Zhu et al., 2022) resize them to 240 × 320.

Table 1: Quantitative comparisons on inverse neural rendering task. Our method outperforms most of the metrics on both object and scene level test dataset.

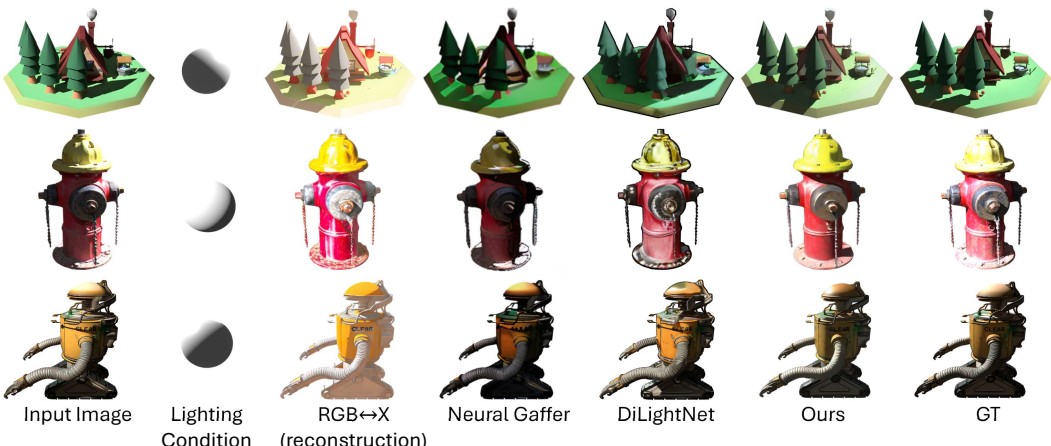

Input Image | Lighting Condition | RGB↔X (reconstruction) | Neural Gaffer | DiLightNet | Ours | GT

Figure 6: Qualitative comparisons with the state-of-the-art object relighting methods.

**Evaluation Protocols.** We randomly select 50 objects from Objaverse (Deitke et al., 2023b;a) and 20 scenes from BlenderKit (BlenderKit, 2024), which are not included in the training dataset. For each object, we rendered images under 6 different lighting conditions and 4 camera views, yielding a total of $4 \times 6 \times 50$ images. For the scene data, we rendered 200 images from various viewpoints.

| Method | Dataset | PSNR↑ | SSIM↑ | LPIPS↓ | Dataset | PSNR↑ | SSIM↑ | LPIPS↓ |
|---|---|---|---|---|---|---|---|---|
| RGB↔X | Object50 | - | - | - | Object50 (reconstruction) | 9.64 | 0.8878 | 0.0783 |
| Neural Gaffer | | 11.72 | 0.9069 | 0.0844 | | 13.69 | 0.9186 | 0.0765 |
| DiLightNet | | 12.21 | 0.8938 | 0.0890 | | 12.72 | 0.9035 | 0.0816 |
| **Ours** | | **14.09** | **0.9211** | **0.0553** | | **15.06** | **0.9332** | **0.0458** |

Table 2: Quantitative comparisons on forward rendering task. Our method outperforms other state-of-the-art methods on object50 test dataset. (reconstruction) means the input image, output image and lighting condition are well aligned, that is, the inputs are identical to the ground-truth outputs.

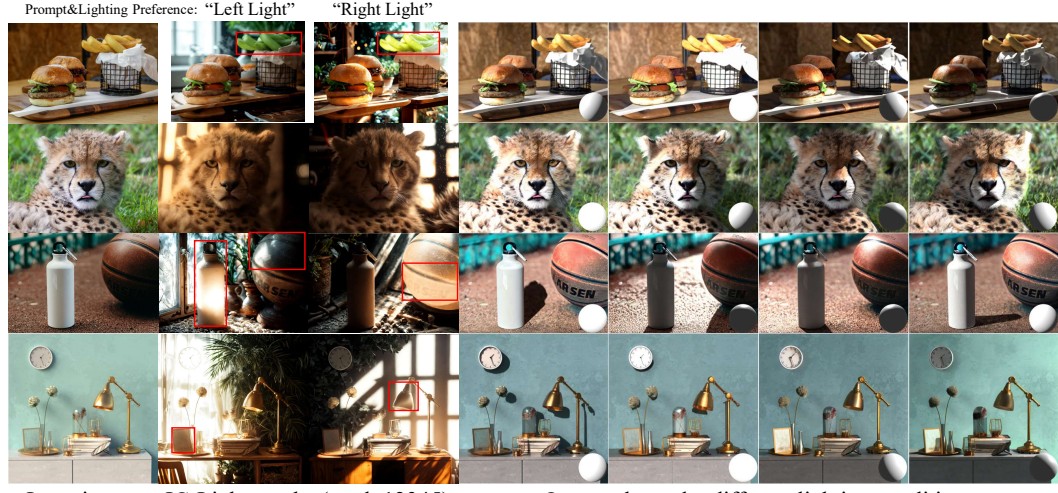

Figure 7: Qualitative comparisons with the state-of-the-art relighting method IC-Light (Zhang et al., 2025) demonstrating the generalization ability of our model on real images.

For all quantitative results in our paper, we set CFG to 1.0 in stage 1 and 1.5 in stage 2, with these values selected empirically. The influence of the CFG scale is further discussed in *Appendix* A.3.

## 5.2 COMPARISONS FOR INVERSE NEURAL RENDERING

For inverse rendering, we compare four state-of-the-art inverse neural rendering methods: Intrinsic-Anything (Chen et al., 2024), Zhu et al. (Zhu et al., 2022), Kocsis et al. (Kocsis et al., 2024b), and RGB↔X (Zeng et al., 2024b), using our test datasets mentioned in Sec. 5.1. We evaluate performance using 3 metrics: PSNR, SSIM (Wang et al., 2004), LPIPS (Zhang et al., 2018).

As in Tab. 1, our model achieves state-of-the-art performance on most metrics. Notably, the resolution of Zhu et al. (2022) and Kocsis et al. (2024b) is relatively low, which may contribute to higher pixel-supervised scores. As in Fig. 5, our method produces more precise and clearer results. For instance, in the first row, our model successfully predicts the albedo of objects reflected in a mirror, whereas other methods incorrectly treat them as part of the mirror surface.

## 5.3 EVALUATIONS OF NEURAL FORWARD RENDERING

As shown in Tab. 2 and Fig. 6, our model outperforms existing object relighting methods under our evaluation setting. Since the second stage of RGB↔X (Zeng et al., 2024b) does not involve environment map control, we report metrics under the reconstruction setting. As observed in the first row, DiLightNet (Zeng et al., 2024a) tends to bake the input shadow into the output. Similarly, in the second and third rows, Neural Gaffer (Jin et al., 2024) also exhibits shadow baking when the shadow covers a large area. In contrast, RGB↔X (Zeng et al., 2024b) is trained only on indoor scenes, resulting in limited generalization capability. In comparison, our method does not suffer from these issues and achieves more faithful relighting results.

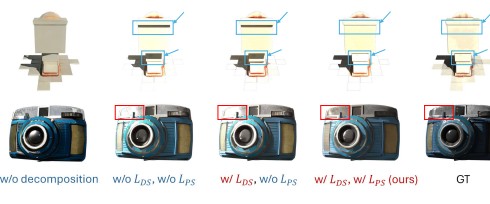

w/o decomposition  w/o $L_{DS}$, w/o $L_{PS}$   w/ $L_{DS}$, w/o $L_{PS}$   w/ $L_{DS}$, w/ $L_{PS}$ (ours)   GT

Figure 8: Qualitative comparisons in the ablation study for the effects of decomposition and the two shading loss functions proposed in our method.(zoom in for a better view)

Table 3: Quantitative comparison between pipelines with and without decomposition, as well as with and without the two shading loss functions proposed in our method. All models are trained for $30k$ iterations to ensure fairness.

| Method | | | Metrics | | |
|---|---|---|---|---|---|
| Decomposition | $L_{DS}$ | $L_{PS}$ | PSNR↑ | SSIM↑ | LPIPS↓ |
| | | | 10.88 | 0.8964 | 0.0712 |
| ✓ | | | 11.95 | 0.9072 | 0.0616 |
| ✓ | ✓ | | 11.97 | 0.9054 | 0.0617 |
| ✓ | | ✓ | 12.18 | 0.9070 | 0.0620 |
| ✓ | ✓ | ✓ | 12.68 | 0.9139 | 0.0572 |

As shown in Fig. 7, our model accurately generates specular highlights, particularly on highly metallic objects such as the desk lamp in the fourth row. Additionally, it successfully produces shadows on the ground or the wall, as seen with the basketball and water cup in the second row. While IC-Light (Zhang et al., 2025) exhibits two main limitations: 1) It fails to preserve the object's albedo and roughness, as highlighted in the red box, where both the object's color and glossiness have changed. 2) It lacks precise lighting control; for instance, in the second row, despite providing a right-light prompt, the illumination still originates from the left.

## 5.4 ABLATION STUDIES

**The Necessity of Image Intrinsic Decomposition.** As shown in Fig. 8 and Tab. 3 (w/o decomposition), we conducted an end-to-end training experiment using only the lighting condition as input and evaluated the results on the object50 test dataset. We observed that without decomposition, as shown in Fig. 8 (bottom row), the model fails to accurately perceive the object's normal, leading to unrealistic shading effects that do not align with the object's true orientation. As shown in Fig. 8 (top row), the albedo varies with lighting changes, further deviating from physically consistent behavior.

**The Effectiveness of the Diffuse Shading Loss and Physical-based Shading Loss.** As shown in Fig. 8 and Tab. 3, we evaluate the effectiveness of the two shading loss functions on the object50 test dataset. Tab. 3 indicates that both losses significantly enhance model performance. Moreover, a comparison between the third and fourth columns in Fig. 8 reveals that the physically-inspired shading loss effectively corrects highlight and shadow directions.

## 6 CONCLUSION

We have introduced a novel neural rendering diffusion-based approach for full-image relighting. To support this, we constructed a dataset containing objects and scenes under diverse lighting conditions. Our method incorporates physical light transport, enabling controllable illumination while preserving the intrinsic physical properties of objects. Experimental results demonstrate the superiority of our approach in predicting physical attributes and the effectiveness and controllability of our relighting results.

## ACKNOWLEDGMENT

This research is supported by the National Research Foundation, Singapore, under its NRF Fellowship Award NRF-NRFF16-2024-0003. This research is also supported by NTU SUG-NAP, as well as cash and in-kind funding from NTU S-Lab and industry partner(s).

## REPRODUCIBILITY STATEMENT

To ensure that all results, including both visual and quantitative ones, are reproducible, we fixed the random seed to 42 for all experiments. For the proposed datasets, a complete description of the data processing steps is provided in Appendix A.1.

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

# A APPENDIX

## A.1 DATASET CONSTRUCTION

To train a model capable of accurately decomposing an input RGB image, we need to collect objects and scenes that contain rich material information. At the object level, we filter objects from Objaverse (Deitke et al., 2023b;a), selecting those with BRDF material properties. For scene data, while some existing datasets Kocsis et al. (2024b); Zeng et al. (2024b) provide intrinsic components, they primarily focus on indoor scenes and lack object masks to exclude translucent or transparent objects, which can hinder accurate material estimation. Moreover, these datasets do not render images under varying lighting conditions—a critical factor for our task. To address these limitations, we curate a scene dataset from an online repository BlenderKit (2024) and render intrinsic components along with diverse lighting conditions for training. The following section details our dataset construction pipeline.

### A.1.1 OBJECT DATA

For object-centric data, we randomly select over 10,000 objects from filtered Objaverse (Deitke et al., 2023b;a) that contain BRDF materials and render them under varying viewpoints and lighting conditions. Each object is rendered with 10 different views and 10 different lighting conditions, resulting in a total of 100 images per object. The first four views are captured at a $70°$ elevation angle from the front, back, left, and right, while the remaining views are randomly sampled.

The lighting direction is randomly sampled. The horizontal angle is uniformly distributed in $[0°, 360°]$, while the vertical angle follows a normal distribution centered around $60°$ with a standard deviation of $40°$, clipped to the range $[20°, 160°]$. This sampling strategy biases the light direction toward the upper hemisphere, reflecting a higher probability of natural top-down lighting. The sunlight strength is randomly sampled in the range $[5, 20]$, with a slight random variation, and is reduced when HDRI lighting is used. A small random ambient light is added to avoid extremely dark areas. In 80% of cases, a point light source is used, while the remaining 20% incorporate both a point light source and an environment map. The environment map is randomly selected from a set of 700 maps sourced from Poly Haven (Haven, 2025).

BlenderProc (Denninger et al., 2023) utilizes Blender's composition layer to generate intrinsic properties. However, we observed that certain albedo values fluctuate with the light source, and for transparent or semi-transparent materials, the composition layer often outputs entirely white or black values, which are inaccurate. To mitigate these issues, we extract all intrinsics directly from Blender's Principled BRDF nodes while simultaneously generating a mask to identify regions that are not rendered using the Principled BRDF model.

To effectively model the lighting of a full image, we propose a novel light representation different from all previous methods (Zhang et al., 2025; Bashkirova et al., 2023; Zeng et al., 2024a). Most existing methods either condition on global HDRI lighting information (Zeng et al., 2024a) or infer lighting from the background to relight the foreground (Zhang et al., 2025; Bashkirova et al., 2023). In contrast, our method constructs the lighting condition using only the light sources directly facing the image, leveraging only the front hemisphere of a gray ball rendered from the HDRI. This representation effectively mitigates the influence of self-luminous objects and built-in light sources (e.g., light from windows) present in the original scene.

Specifically, we position a camera at the same location as the input image and render a gray ball with 50% roughness in Blender under the given lighting condition. The rendered image is saved as an RGB image, $L \in \mathbb{R}^{C \times H \times W}$, where the sphere is tangent to the image boundaries.

### A.1.2 SCENE DATA

For scene data, we downloaded 400 scenes from BlenderKit (BlenderKit, 2024), consisting of 300 indoor and 100 outdoor environments. After manual filtering, we retained 300 scenes by removing those containing excessive special effects, such as fog and rain. The rendering process for these scenes is largely similar to that of object data, with slight modifications in view selection. Specifically, we applied parallel translations within a range of [-0.5, 0.5] along the planar direction perpendicular to the camera orientation, random forward or backward translations within a range of [-0.2,

0.5] along the camera direction, and minor rotations within a 5° elevation angle. These adjustments prevent the camera from penetrating walls or moving outside the scene boundaries.

The lighting conditions were generated by randomly adding a point light source to the original scene illumination setup. The additional point light was placed behind the camera, with an angular deviation of [-30°, 30°] relative to the camera's viewing direction. Notably, we observed that normal map rendering varies across different Blender versions. To ensure consistency, we filtered scenes suitable for training and manually specified the Blender version used for each scene.

We construct the Object50 and Scene200 datasets using the same procedure as the training set construction. To prevent data leakage, the object IDs and scenes in the evaluation sets are strictly disjoint from those in the training set.

## A.2 Validating Physical Consistency in Relighting

As shown in Fig. 9, when we modify the normal map by flipping the R channel, which corresponds to inverting the x-coordinate in camera space, we observe that under the same lighting condition, the generated illumination and shadows change accordingly.

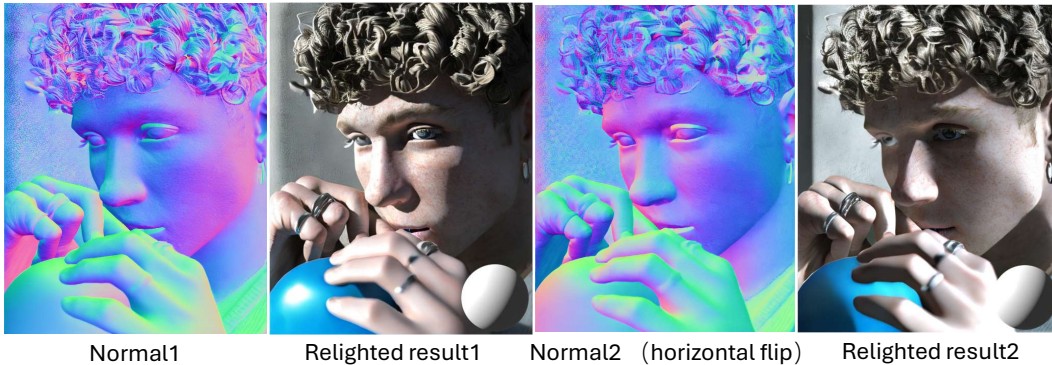

| Normal1 | Relighted result1 | Normal2 (horizontal flip) | Relighted result2 |

Figure 9: Our stage-2 model successfully perceives changes in light direction caused by variations in surface normals.

## A.3 Ablation Studies on the Weight of Classifier-Free Guidance (CFG)

As shown in Tab. 4, we analyze the impact of different CFG weights on the results using the object test dataset. We observe that increasing the CFG value leads to outputs that are more closely aligned with the conditioning input. For instance, in the case of albedo estimation, a larger CFG encourages the model to preserve more details from the input image, inadvertently baking shading effects into the albedo, which degrades the quality of the results. To mitigate this issue, we set CFG to 1.0 in the first stage, that is, disable CFG, and selected a relatively small CFG value in the second stage to ensure that the generated results appear more physically accurate.

We provide qualitative comparisons of different CFG weights in Fig. 10. Consistent with our observations in the main paper, increasing the CFG weight causes the model to produce intrinsics that deviate from physical properties, indicating a stronger resemblance to the condition image. As a result, higher CFG values lead to more shading baked into the albedo, while the normal maps become sharper but exhibit more artifacts.

| Element | Scale | Albedo | | | Normal | | | Roughness | | | Metallic | | | Average | | |
|---|---|---|---|---|---|---|---|---|---|---|---|---|---|---|---|---|
| | | PSNR↑ | SSIM↑ | LPIPS↓ | PSNR↑ | SSIM↑ | LPIPS↓ | PSNR↑ | SSIM↑ | LPIPS↓ | PSNR↑ | SSIM↑ | LPIPS↓ | PSNR↑ | SSIM↑ | LPIPS↓ |
| CFG | 1.0 | **22.09** | **0.9200** | **0.0562** | 24.97 | 0.9217 | **0.0531** | **21.09** | **0.8961** | **0.0736** | **13.96** | 0.8357 | **0.1216** | **20.53** | **0.8934** | **0.0761** |
| | 1.5 | 21.42 | 0.9153 | 0.0576 | 24.94 | **0.9231** | 0.0509 | 19.28 | 0.8835 | 0.0794 | 13.54 | **0.8364** | 0.1228 | 19.80 | 0.8896 | 0.0777 |
| | 2.0 | 20.56 | 0.9053 | 0.0637 | 24.74 | 0.9207 | 0.0522 | 18.03 | 0.8721 | 0.0863 | 13.42 | 0.8358 | 0.124 | 19.19 | 0.8835 | 0.0816 |
| | 2.5 | 19.79 | 0.8966 | 0.0691 | 24.45 | 0.9181 | 0.0538 | 17.27 | 0.8645 | 0.0916 | 13.4 | 0.8341 | 0.1263 | 18.73 | 0.8783 | 0.0852 |

Table 4: Ablation studies of the influence of the Classifier-Free Guidance (CFG).

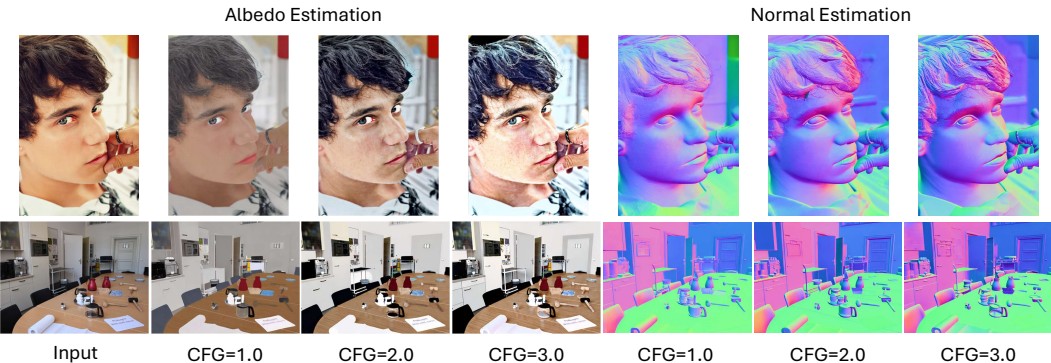

Figure 10: Qualitative comparisons on different CFG weights. Higher CFG values lead to more shading baked into the albedo, while the normal maps become sharper but exhibit more artifacts.

## A.4 ATTRIBUTION OF PERFORMANCE GAINS

As shown in Tab. 5, we reproduce RGB↔X (Zeng et al., 2024b) following the training details in their paper and retrained RGB↔X on the same dataset used by our method. Even after retraining RGB↔X on our dataset, our method consistently achieves better performance, demonstrating that the methodological design itself also contributes to the performance gain.

As shown in Tab. 6, we conducted the additional evaluation on a representative subset of our dataset to show that the advantage is not only gained from the dataset but indeed benefits from the design of our physics-inspired module. Specifically, we retrained Neural Gaffer (Jin et al., 2024) on a subset, which is part of our dataset with both the grey ball and environment maps rendered, for 80k iterations following the original protocol in its paper. We also retrained our framework on the same subset for 30k iterations. Despite the shorter training schedule, our model still outperforms Neural Gaffer on this shared training set, demonstrating that the performance gains are attributable not only to the dataset but also to the effectiveness of our proposed architecture. This also verifies that our physics-inspired loss, as a regularization term, guides the model to converge more quickly toward physically plausible principles.

| Dataset | Method | Albedo | | | Normal | | | Roughness | | | Metallic | | | Average | | |
|---|---|---|---|---|---|---|---|---|---|---|---|---|---|---|---|---|
| | | PSNR↑ | SSIM↑ | LPIPS↓ | PSNR↑ | SSIM↑ | LPIPS↓ | PSNR↑ | SSIM↑ | LPIPS↓ | PSNR↑ | SSIM↑ | LPIPS↓ | PSNR↑ | SSIM↑ | LPIPS↓ |
| Object50 | RGB↔X (Zeng et al., 2024b) | 20.09 | 0.9128 | 0.0737 | 22.78 | 0.9039 | 0.0793 | 18.43 | **0.9061** | 0.0752 | **15.97** | **0.8404** | 0.1226 | 19.32 | 0.8908 | 0.0877 |
| | RGB↔X (retrained) (Zeng et al., 2024b) | 19.51 | 0.9071 | 0.0726 | 22.34 | 0.8908 | 0.0846 | 20.47 | 0.8876 | 0.0800 | 15.19 | 0.8309 | 0.1227 | 19.38 | 0.8791 | 0.0900 |
| | Ours | **22.09** | **0.9200** | **0.0562** | **24.97** | **0.9217** | **0.0531** | **21.09** | 0.8961 | **0.0736** | 13.96 | 0.8357 | **0.1216** | **20.53** | **0.8934** | **0.0761** |
| Scene200 | RGB↔X (Zeng et al., 2024b) | 14.07 | 0.6955 | 0.2441 | 14.59 | 0.6824 | 0.3278 | 11.14 | 0.6187 | 0.3086 | **7.20** | **0.2217** | 0.6764 | 11.75 | 0.5546 | 0.3892 |
| | RGB↔X (retrained) (Zeng et al., 2024b) | 13.16 | 0.6659 | 0.2802 | 14.68 | 0.6025 | 0.3876 | 12.08 | 0.5453 | 0.3977 | 6.86 | 0.1928 | **0.6748** | 11.70 | 0.5016 | 0.4351 |
| | Ours | **14.46** | **0.7025** | **0.2248** | **16.26** | **0.7208** | **0.2768** | **14.05** | **0.6348** | **0.2980** | 7.02 | 0.2001 | 0.6966 | **12.95** | **0.5646** | **0.3741** |

Table 5: Quantitative comparisons on inverse neural rendering task. Our method outperforms most of the metrics on both object and scene level test dataset.

| Method | PSNR↑ | SSIM↑ | LPIPS↓ |
|---|---|---|---|
| Neural Gaffer (Jin et al., 2024) | 10.31 | 0.8909 | 0.0937 |
| Ours | **11.74** | **0.9060** | **0.0637** |

Table 6: Quantitative comparisons of the forward neural rendering task.

## A.5 LIGHT INVARIANCE OF THE RECONSTRUCTION LOSS

Here, we provide additional details and discussion regarding the reconstruction loss introduced in Sec. 4.3.2 of the main paper. We use the feature from the last hidden layer of the DINOv2-base model for reconstruction. This layer was selected empirically based on performance in our experiments.

As shown in Fig.11, the features from this layer are more sensitive to texture while being largely invariant to lighting. For example, as illustrated in the table on the right of Fig.11, when the lighting conditions differ but the scene texture remains the same, the feature similarity remains above 97%.

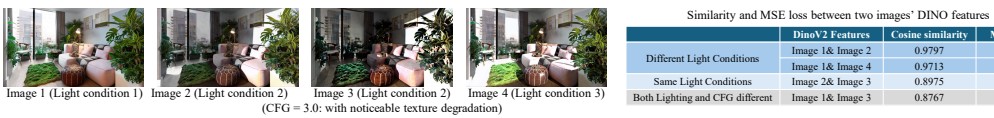

| | | Similarity and MSE loss between two images' DINO features | |
| --- | --- | --- | --- |
| | **DinoV2 Features** | **Cosine similarity** | **MSE Loss** |
| Different Light Conditions | Image 1& Image 2 | 0.9797 | 0.0510 |
| | Image 1& Image 4 | 0.9713 | 0.0769 |
| Same Light Conditions | Image 2& Image 3 | 0.8975 | 0.2698 |
| Both Lighting and CFG different | Image 1& Image 3 | 0.8767 | 0.3151 |

Figure 11: Light invariance of the DINO reconstruction loss.

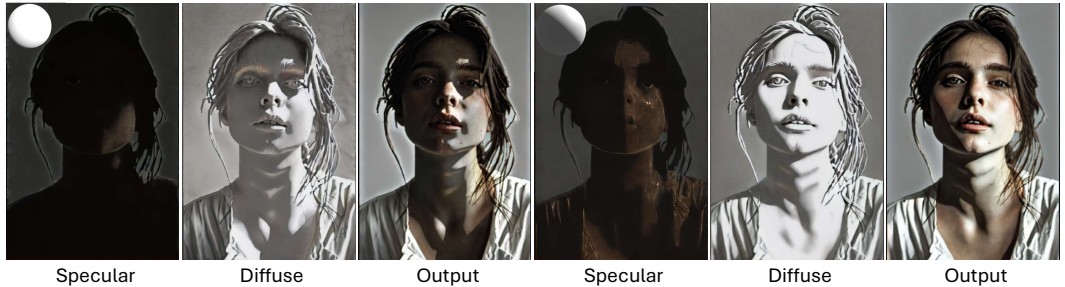

| Specular | Diffuse | Output | Specular | Diffuse | Output |

Figure 12: More Explanation on the Physical-based Shading Loss.

In contrast, when the lighting is held constant but the scene undergoes degradation (simulated by using a high CFG value), the similarity drops and the MSE loss increases significantly.

Therefore, we leverage features from this layer to construct a light-invariant reconstruction loss that enforces scene consistency under varying lighting conditions and helps reduce artifacts.

### A.6 MORE EXPLANATION ON THE PHYSICAL-BASED SHADING LOSS

Without the Physical-based Shading Loss, when encountering highly out-of-domain data, the relighting output tends to remain consistent with the input and does not change in response to variations in the diffuse and specular components, even if the generated diffuse and specular maps are correct and vary with lighting conditions. However, after introducing the Physical-based Shading Loss, as shown in Fig. 12, the final output follows the diffuse and specular maps and becomes consistent with them.

### A.7 MORE EXPLANATION ON THE BATCH-AWARE SELF-ATTENTION

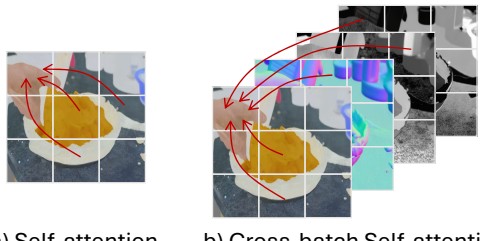

a) Self-attention    b) Cross-batch Self-attention

Figure 13: Illustration of standard self-attention and the cross-batch self-attention used in our method.

As shown in Fig. 13, our batch-aware self-attention enforces stronger consistency among albedo, normal, roughness, and metallic predictions. In the cross-batch self-attention layers, the queries, keys, and values are computed as follows:

$$q_a = W_q \hat{z}^a, k_a = W_k(\hat{z}^a \oplus \hat{z}^n \oplus \hat{z}^r \oplus \hat{z}^m), v_a = W_v(\hat{z}^a \oplus \hat{z}^n \oplus \hat{z}^r \oplus \hat{z}^m)$$

$$q_n = W_q \hat{z}^n, k_n = W_k(\hat{z}^n \oplus \hat{z}^a \oplus \hat{z}^r \oplus \hat{z}^m), v_n = W_v(\hat{z}^n \oplus \hat{z}^a \oplus \hat{z}^r \oplus \hat{z}^m)$$

$$q_r = W_q \hat{z}^r, k_r = W_k(\hat{z}^r \oplus \hat{z}^a \oplus \hat{z}^n \oplus \hat{z}^m), v_r = W_v(\hat{z}^r \oplus \hat{z}^a \oplus \hat{z}^n \oplus \hat{z}^m)$$

$$q_m = W_q \hat{z}^m, k_m = W_k(\hat{z}^m \oplus \hat{z}^a \oplus \hat{z}^r \oplus \hat{z}^n), v_m = W_v(\hat{z}^m \oplus \hat{z}^a \oplus \hat{z}^r \oplus \hat{z}^n)$$

where, $\hat{z}^a, \hat{z}^n, \hat{z}^r, \hat{z}^m$ are latent albedo, normal, roughness and metallic embeddings in transformer blocks. The operator $\oplus$ denotes concatenation, and $W_q$, $W_k$ and $W_v$ are the projection matrices used to compute the query, key, and value embeddings. The cross-batch features are $Att(q_i, k_i, v_i), i = \{a, n, r, m\}$, where $Att()$ denotes the standard softmax attention.

## A.8 MORE RESULTS

### A.8.1 DECOMPOSITION AND RELIGHTING

As shown in Fig. 25, 26, 27 and 28, we provide more results for both of our inverse and forward rendering pipeline on real images. Our results output reasonable intrinsic components and controllable relighted images under different lighting conditions.

## A.9 MORE COMPARISONS

### A.9.1 QUALITATIVE ABLATIONS ON OUT-OF-DISTRIBUTION IMAGE RELIGHTING

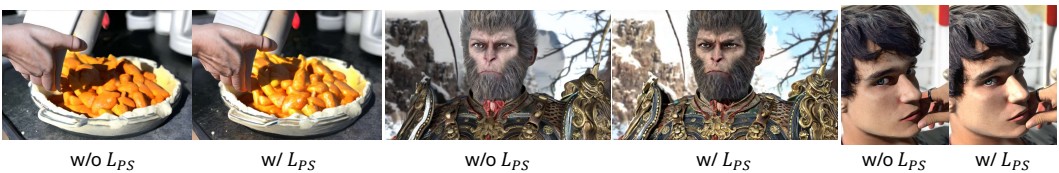

w/o $L_{PS}$  w/ $L_{PS}$    w/o $L_{PS}$  w/ $L_{PS}$    w/o $L_{PS}$  w/ $L_{PS}$

Figure 14: Qualitative comparisons on out-of-distribution inputs, with and without the physics-inspired loss functions. Incorporating the physics-inspired loss produces more realistic and detailed results, demonstrating its contribution to improved generalization.

Here we include qualitative comparisons on out-of-distribution inputs, with and without the physics-inspired loss functions. As shown in Fig. 14, incorporating the physics-inspired loss produces more realistic and detailed results, demonstrating its contribution to improved generalization.

### A.9.2 MORE QUALITATIVE COMPARISONS ON REAL IMAGE RELIGHTING

As shown in Fig. 15, we provide additional qualitative comparisons on real image relighting. We compare our method with two widely used approaches, IC-Light (Zhang et al., 2025) and ChatGPT-4o (OpenAI, 2025). As observed in Fig. 15, both IC-Light and ChatGPT-4o struggle to preserve the original foreground albedo. Notably, the monkey appears aged, and the bicycle color is inconsistent with the input image. While ChatGPT-4o produces relatively natural results in the overall scene, it fails to accurately control the lighting direction. For example, even when instructed to apply lighting from the right, it still illuminates from the left. This suggests that incorporating chain-of-thought (CoT) reasoning might be necessary to guide and refine its output through iterative correction. In contrast, our method consistently produces relighting results under precisely controlled lighting conditions while effectively preserving the intrinsic properties of the original images.

### A.9.3 QUALITATIVE COMPARISONS ON SCENE IMAGE RELIGHTING

As shown in Fig. 16, we provide qualitative comparisons on scene image relighting with the state-of-the-art methods RGB↔X(Zeng et al., 2024b), Neural Gffer (Jin et al., 2024), DiLightNet (Zeng et al., 2024a).

### A.9.4 INTRINSIC DECOMPOSITION ON REAL IMAGE

As shown in Fig. 17 and 18, we provide qualitative comparisons with RGB↔X (Zeng et al., 2024b) on real image intrinsic decomposition task. As shown in Fig. 21,22,23 and 24, we also provide extended versions of Fig. 5 from the main paper.

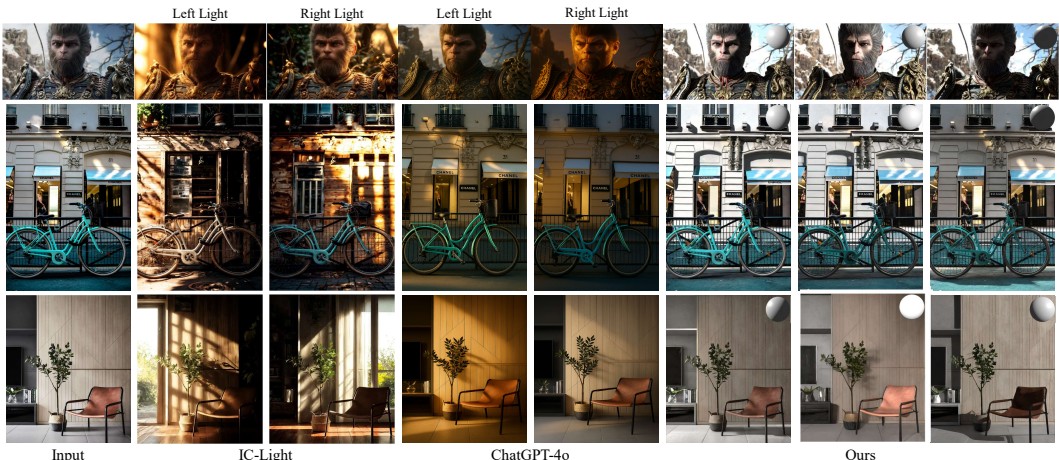

Figure 15: Additional qualitative comparisons on real image relighting. Both IC-Light and ChatGPT-4o struggle to preserve the original foreground albedo, and ChatGPT-4o fails to accurately follow the specified lighting conditions. In contrast, our method consistently produces relighting results under precisely controlled lighting directions while effectively preserving the intrinsic properties of the original images.

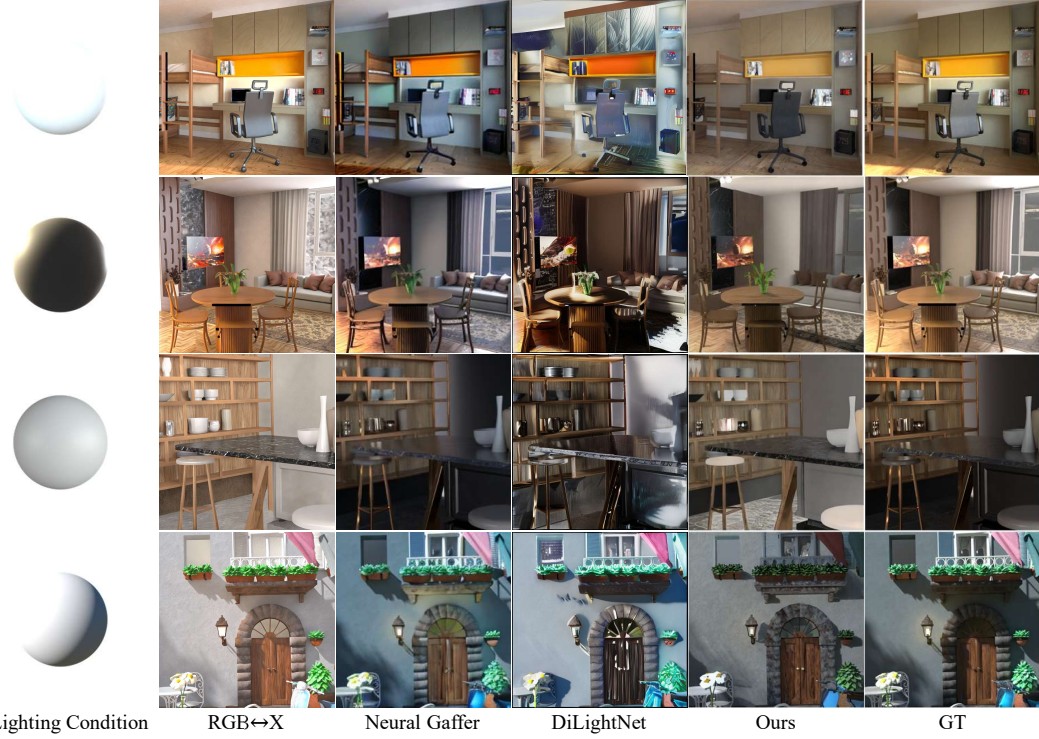

Figure 16: Qualitative comparisons on scene image relighting.

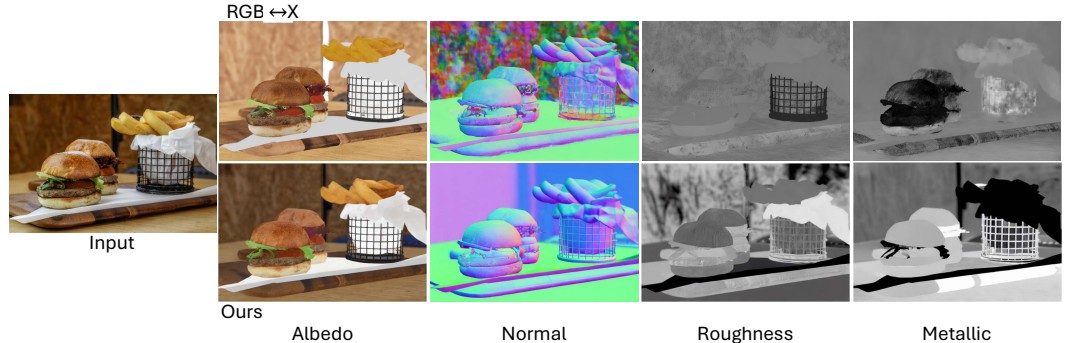

Figure 17: Qualitative comparison with RGB↔X on real image. Our method produces more detailed and physically consistent intrinsics.For instance, our model successfully predicts the high metallicity of the iron basket in the picture.

### A.9.5 NORMAL ESTIMATION

As shown in Fig. 20, our method can generate more reasonable and detailed normal map even compared with the state-of-the-art normal estimation methods, DSINE (Bae & Davison, 2024) and GeoWizard (Fu et al., 2024).

We also evaluated the predicted surface normals using the Mean Angular Error (MAE) metric. Quantitative comparisons based on MAE are reported in Tab. 7.

### A.9.6 EXTENDED TEST DATASETS

We have expanded our test split from 50 objects / 20 scenes to 500 objects / 36 scenes, that is, obj500 contains 500 objects resulting in 12000 images, and the updated quantitative results are reported in Tab. 8.

### A.10 ADDITIONAL SETUP FOR TRAINING.

Since the loss function in stage two requires loading the VAE decoder, which significantly increases memory consumption, we employed DeepSpeed (Rasley et al., 2020) with ZeRO (Rajbhandari et al., 2020) stage 2 and mixed precision set to 'bf16' to optimize memory usage during training.

### A.11 DISSCUSIONS

### A.11.1 FRAMEWORK DISTINCTIONS FROM PRIOR METHODS

Our framework design differs from RGB↔X (Zeng et al., 2024b) and DiffusionRenderer (Liang et al., 2025). In the inverse neural rendering stage, both prior methods generate one intrinsic component at a time using prompt-based textual inputs, resulting in lower efficiency and limited consistency across outputs. In contrast, our method predicts multiple intrinsic components simultaneously, facilitated by a cross-batch self-attention mechanism that allows information exchange between different intrinsics. Specifically, following Wonder3D (Long et al., 2024), we extend standard self-

| Method | Object50 | | | Scene200 | | |
|---|---|---|---|---|---|---|
| | 11.25° ↑ | 22.5° ↑ | 30° ↑ | 11.25° ↑ | 22.5° ↑ | 30° ↑ |
| Kocsis et al.* (Kocsis et al., 2024b) | 34.8 | 60.7 | 72.4 | 59.4 | 74.2 | 80.7 |
| Zhu et al.* (Zhu et al., 2022) | 10.6 | 31.3 | 44.7 | 43.1 | 60.1 | 68.6 |
| RGB↔X (Zeng et al., 2024b) | 19.5 | 42.5 | 53.8 | 44.5 | 60.7 | 67.2 |
| Ours | 36.0 | 64.0 | 74.1 | 52.1 | 68.3 | 75.8 |

Table 7: Quantitative comparisons of normal estimation based on MAE metric.

| Dataset | Method | Albedo | | | Normal | | | Roughness | | | Metallic | | | Average | | |
|---|---|---|---|---|---|---|---|---|---|---|---|---|---|---|---|---|
| | | PSNR↑ | SSIM↑ | LPIPS↓ | PSNR↑ | SSIM↑ | LPIPS↓ | PSNR↑ | SSIM↑ | LPIPS↓ | PSNR↑ | SSIM↑ | LPIPS↓ | PSNR↑ | SSIM↑ | LPIPS↓ |
| Object500 | IntrinsicAnything (Chen et al., 2024) | 15.78 | 0.8535 | 0.1191 | - | - | - | - | - | - | - | - | - | - | - | - |
| | Kocsis et al.* (Kocsis et al., 2024b) | 19.85 | 0.8997 | 0.0852 | 24.93 | 0.9350 | 0.0665 | 16.75 | 0.8877 | 0.0804 | 14.14 | 0.8435 | 0.1245 | 18.92 | 0.8915 | 0.0892 |
| | Zhu et al.* (Zhu et al., 2022) | 19.17 | 0.8681 | 0.0933 | 21.60 | 0.8817 | 0.0903 | 15.98 | 0.8500 | 0.1082 | 12.23 | 0.8140 | 0.1572 | 17.25 | 0.8535 | 0.1123 |
| | RGB↔X (Zeng et al., 2024b) | 19.86 | 0.9015 | 0.0807 | 22.78 | 0.9089 | 0.0862 | 17.27 | 0.8881 | 0.0773 | 16.20 | 0.8408 | 0.1261 | 19.03 | 0.8848 | 0.0926 |
| | Ours | 20.47 | 0.9039 | 0.0661 | 24.61 | 0.9261 | 0.0594 | 19.51 | 0.8859 | 0.0793 | 14.90 | 0.8384 | 0.1199 | 19.87 | 0.8886 | 0.0812 |
| Scene36 | IntrinsicAnything (Chen et al., 2024) | 13.10 | 0.6175 | 0.3655 | - | - | - | - | - | - | - | - | - | - | - | - |
| | Kocsis et al.* (Kocsis et al., 2024b) | 14.49 | 0.6793 | 0.2854 | 18.37 | 0.7578 | 0.2634 | 9.92 | 0.5988 | 0.3269 | 5.34 | 0.2309 | 0.5901 | 12.03 | 0.5667 | 0.3665 |
| | Zhu et al.* (Zhu et al., 2022) | 13.61 | 0.6408 | 0.2630 | 16.32 | 0.6694 | 0.2648 | 9.30 | 0.5262 | 0.3301 | 5.08 | 0.2076 | 0.6037 | 11.08 | 0.5110 | 0.3654 |
| | RGB↔X (Zeng et al., 2024b) | 13.81 | 0.6807 | 0.2629 | 14.64 | 0.6675 | 0.3424 | 10.92 | 0.6078 | 0.3190 | 7.11 | 0.2151 | 0.6749 | 11.62 | 0.5428 | 0.3998 |
| | Ours | 14.10 | 0.6852 | 0.2428 | 16.24 | 0.7122 | 0.2757 | 13.60 | 0.6151 | 0.3129 | 6.97 | 0.2021 | 0.6915 | 12.73 | 0.5537 | 0.3807 |

* indicates that the resolution differs from the original resolution 768 × 768 of our test dataset. Kocsis et al.(Kocsis et al., 2024b) resize images to 480 × 640, while Zhu et al.(Zhu et al., 2022) resize them to 240 × 320.

Table 8: Quantitative comparisons on inverse neural rendering task. Our method outperforms most of the metrics on both object and scene level test dataset.

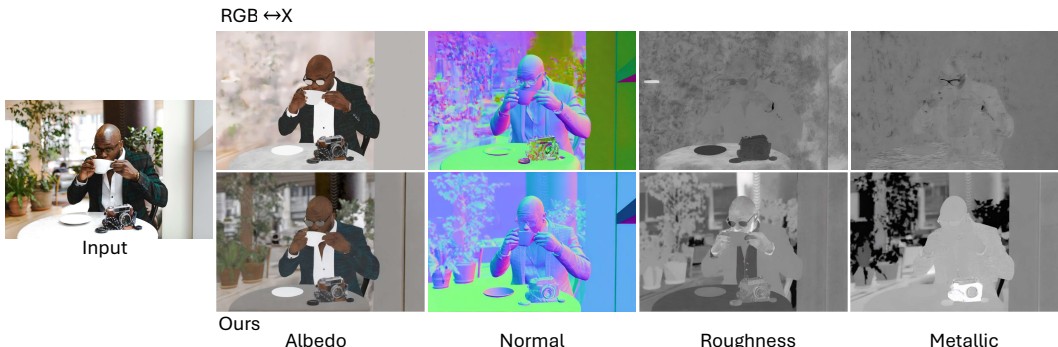

Figure 18: Qualitative comparison with RGB↔X on real images. Our method produces more detailed and physically consistent intrinsics.

attention layers to be global-aware, enabling communication across batches. This design improves both efficiency and consistency among predicted intrinsics.

In the neural forward rendering stage, the intrinsic components obtained from the first stage are further injected as conditions through cross-batch concatenation and the same cross-batch self-attention mechanism. Note that, our physics-inspired loss function is non-trivial. The physics-based loss serves as an efficient learning mechanism, which regularize the training dynamics into a physical plausible landscape, allowing the network to converge faster and learn correct light transport while training on less data and compute, compared to the two mentioned works. The ablation study provided in Tab. 3 of the main paper and the study provided in Sec. A.4 prove the effectiveness of our loss design. These design choices further differentiate our approach from the aforementioned methods.

## A.12 LIMITATION

### A.12.1 SPATIAL MISALIGNMENT BETWEEN LATENT AND RGB REPRESENTATIONS

We conducted a diagnostic experiment where we encoded an image into the VAE latent space, masked out part of the latent representation, and then decoded it back to the RGB space. If the latent space were strictly pixel-aligned with the RGB space, the unmasked regions should reconstruct cleanly, unaffected by the masked areas. However, as shown in Fig. 19, we observed that even within the unmasked rectangular region, the reconstruction showed color shifts and blurring artifacts, not only at the boundaries but across the entire retained region.

This suggests that the latent representation is not strictly pixel-aligned: high-frequency information is not localized spatially in a one-to-one correspondence with the input pixels. As a result, masking operations in the latent space can inadvertently corrupt seemingly unrelated regions during decoding. This misalignment introduces a subtle but systematic error when computing masked losses in latent space, which can propagate into the final output during training. Although the resulting artifacts may not be highly visible, they nonetheless degrade overall performance. As mentioned in the main paper, we have not yet found a better alternative to mitigate this issue. This could be a limitation for further improvement.

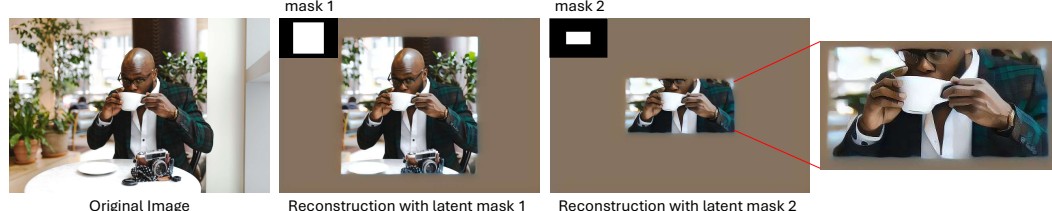

Figure 19: **Latent-space masking diagnostic.** From left to right: original image; decoded reconstruction after masking the VAE latent with Mask 1; decoded reconstruction after masking the VAE latent with Mask 2; zoomed-in reconstruction details.

### A.12.2 LIMITATIONS ON IMAGE INTRINSIC DECOMPOSITION

Image intrinsic decomposition is an ambiguous task with multiple possible solutions, making inaccuracies in the first-stage results inevitable. While our second-stage model has some tolerance for such errors, severe inaccuracies can lead to distortions in the final rendering or deviations from the original image.

The predicted metallic maps may sometimes appear inconsistent with the ground-truth labels. For example, the wooden floor shown in the last row of Fig. 5 in our scene is predicted as highly metallic despite being a non-metallic material. This behavior can be explained by the limitations of the current material representation.

In Blender's Principled BSDF model, in addition to the primary attributes such as albedo, normal, metallic, and roughness, there are many other factors that influence surface reflectance, such as `sheen`, `coat`, and the index of refraction (IOR). For example, non-metallic surfaces with high IOR values can still exhibit strong specular reflections. Since our model does not explicitly model these additional properties, their effects may be implicitly absorbed into the predicted metallic values. This does not imply that the model failed to learn the correct behavior; rather, it reflects the limitation that certain reflective factors, like IOR, are not part of the model's representation space. As shown in the last row in Fig. 5 of our paper, the floor is predicted as metallic, while in the original Blender scene its IOR is set to 1.450. Without modeling IOR explicitly, its reflective effect is captured by the metallic prediction.

A possible improvement is to extend the material representation to explicitly incorporate IOR or train an explicit residual prediction to enable broader material support without losing physical interpretability of the other outputs, which may further enhance accuracy in distinguishing between different types of unformulated materials.

### A.12.3 LIMITATIONS ON RELIGHTING

When a wall is present behind or to the sides, the front-hemisphere light sources may be obstructed, leading to results that do not align with the expected lighting conditions.

Under our current setting and design, it is not possible to modify or add visible light emitters in the rear hemisphere of the environment light. It is out of our current scope. This is because our illumination conditioning is restricted to the frontal hemisphere and does not alter the lighting already present in the rear hemisphere. We view this as a more challenging setting and plan to explore it further in future work.

### A.13 BOARDER IMPACTS

**Positive impact.** This work presents a physically grounded relighting framework based on intrinsic decomposition and environment-aware light modeling. The proposed method enables controllable and high-fidelity relighting of objects and scenes, which has potential applications in computer graphics, augmented/virtual reality (AR/VR), digital content creation, and scene editing. On the positive side, our technique may facilitate more realistic rendering and editing tools for creatives, improve lighting consistency in AR/VR applications, and support visual restoration or data augmentation tasks in computer vision.

**Potential negative impact.** There exists a potential risk of misuse, such as manipulating visual evidence or contributing to deceptive media (e.g., by relighting real-world scenes to imply different environments or time-of-day). To mitigate such risks, we plan to release our model and dataset under a research-only license with appropriate usage guidelines.

### A.14 THE USE OF LLM

Large Language Models (LLMs) were used solely for minor grammar correction and stylistic polishing of the manuscript text. They were not involved in the design of the methodology, execution of experiments, analysis of results, or any other aspect of the scientific contribution.

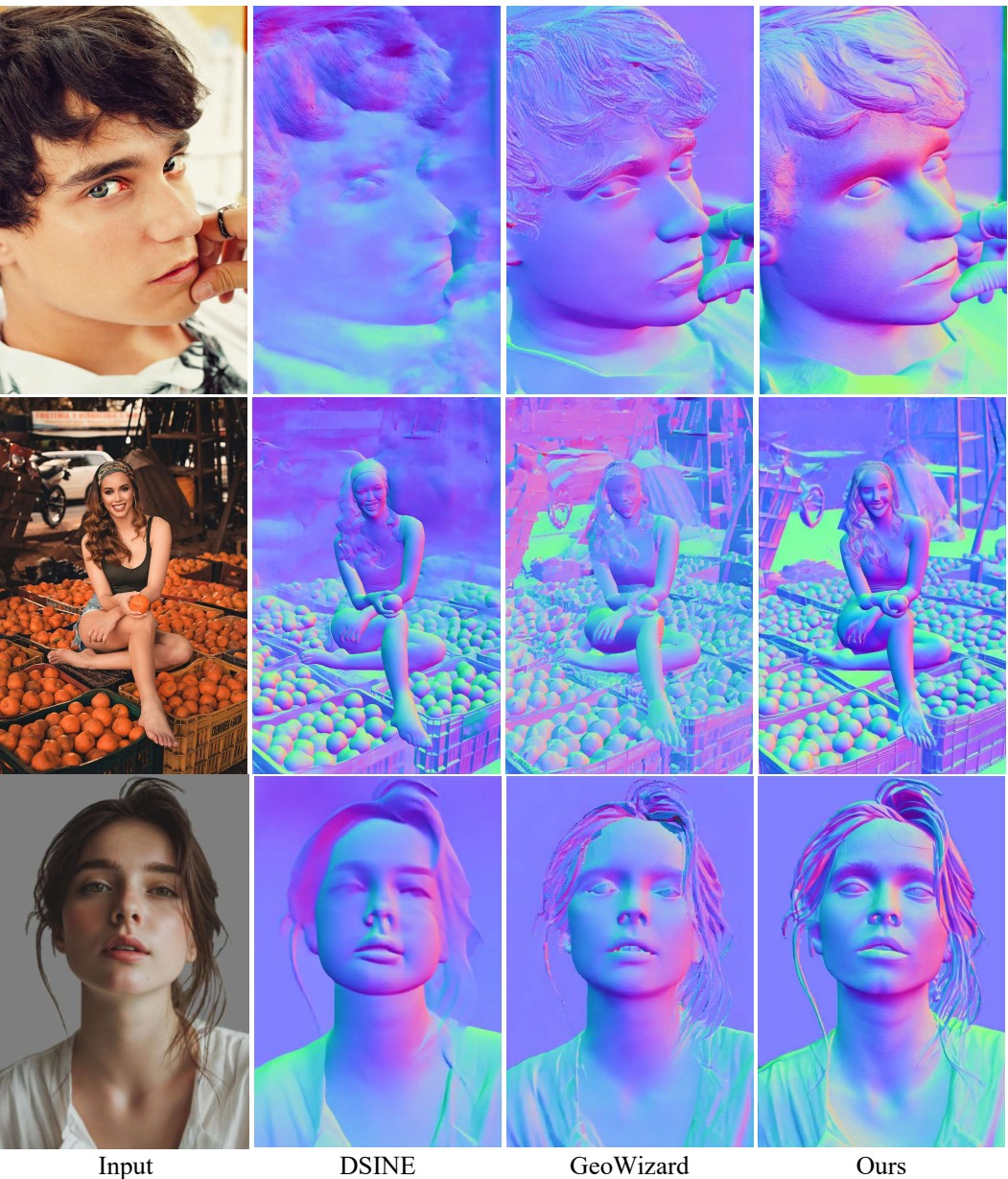

|       Input       |       DSINE       |     GeoWizard     |        Ours       |

Figure 20: Comparisons of the generalization ability with state-of-the-art methods, DSINE (Bae & Davison, 2024) and GeoWizard (Fu et al., 2024) in normal estimation task on real images. Our method can generate more reasonable and detailed normal maps.

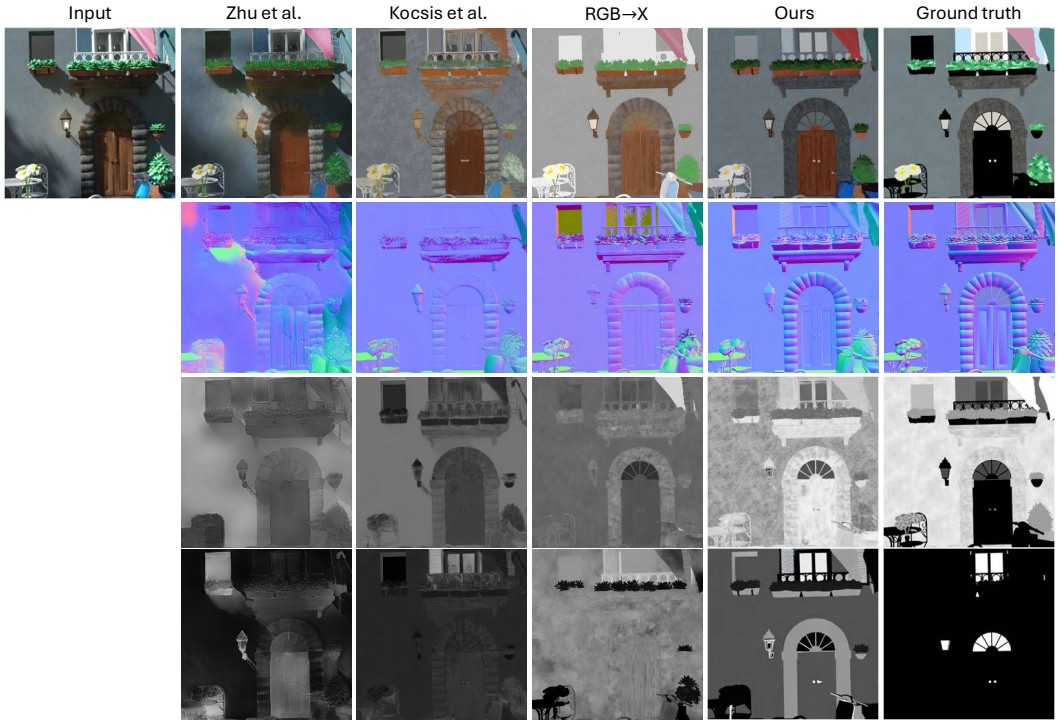

*Part of the black regions in the ground truth represent masked areas, indicating that the shader used in these regions is not based on the Principled BRDF model and has therefore been masked. **(top to button: Albedo, Normal, Roughness, Metallic)**

Figure 21: Qualitative comparisons on our Scene200 test dataset. From top to bottom, the results correspond to albedo, normal, roughness, and metallic comparisons. Our method produces more detailed and accurate results, even in specular reflection regions.

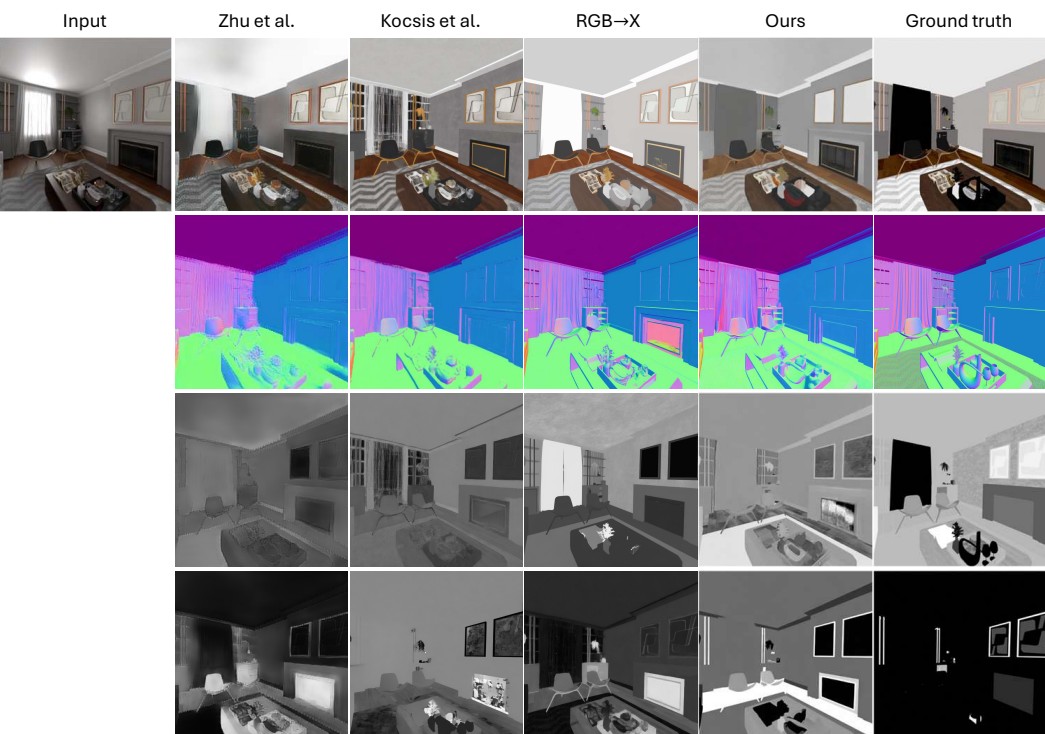

*Part of the black regions in the ground truth represent masked areas, indicating that the shader used in these regions is not based on the Principled BRDF model and has therefore been masked. **(top to button: Albedo, Normal, Roughness, Metallic)**

Figure 22: Qualitative comparisons on our Scene200 test dataset. From top to bottom, the results correspond to albedo, normal, roughness, and metallic comparisons. Our method produces more detailed and accurate results, even in specular reflection regions.

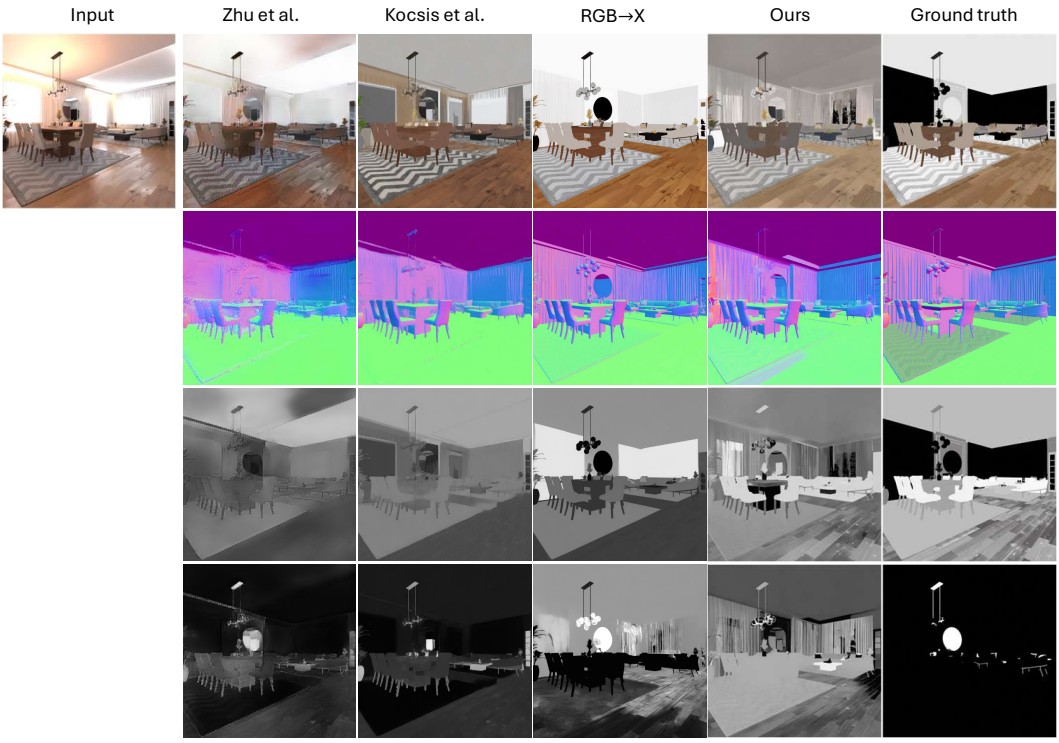

*Part of the black regions in the ground truth represent masked areas, indicating that the shader used in these regions is not based on the Principled BRDF model and has therefore been masked. **(top to button: Albedo, Normal, Roughness, Metallic)**

Figure 23: Qualitative comparisons on our Scene200 test dataset. From top to bottom, the results correspond to albedo, normal, roughness, and metallic comparisons. Our method produces more detailed and accurate results, even in specular reflection regions.

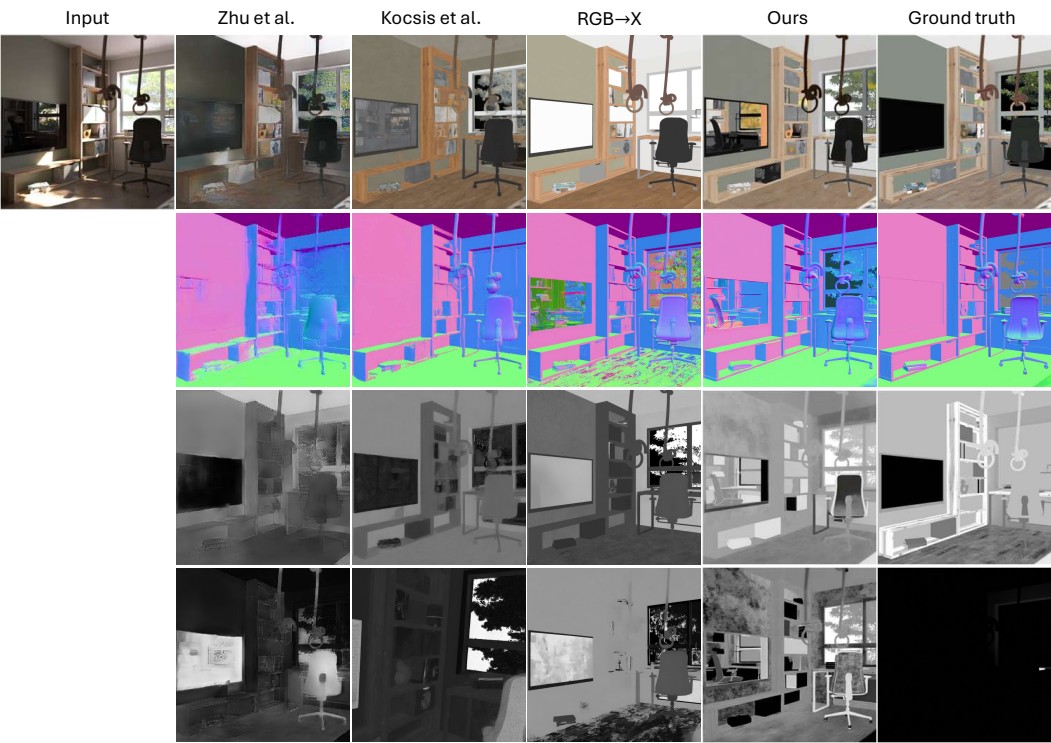

*Part of the black regions in the ground truth represent masked areas, indicating that the shader used in these regions is not based on the Principled BRDF model and has therefore been masked. **(top to button: Albedo, Normal, Roughness, Metallic)**

Figure 24: Qualitative comparisons on our Scene200 test dataset. From top to bottom, the results correspond to albedo, normal, roughness, and metallic comparisons. Our method produces more detailed and accurate results, even in specular reflection regions.

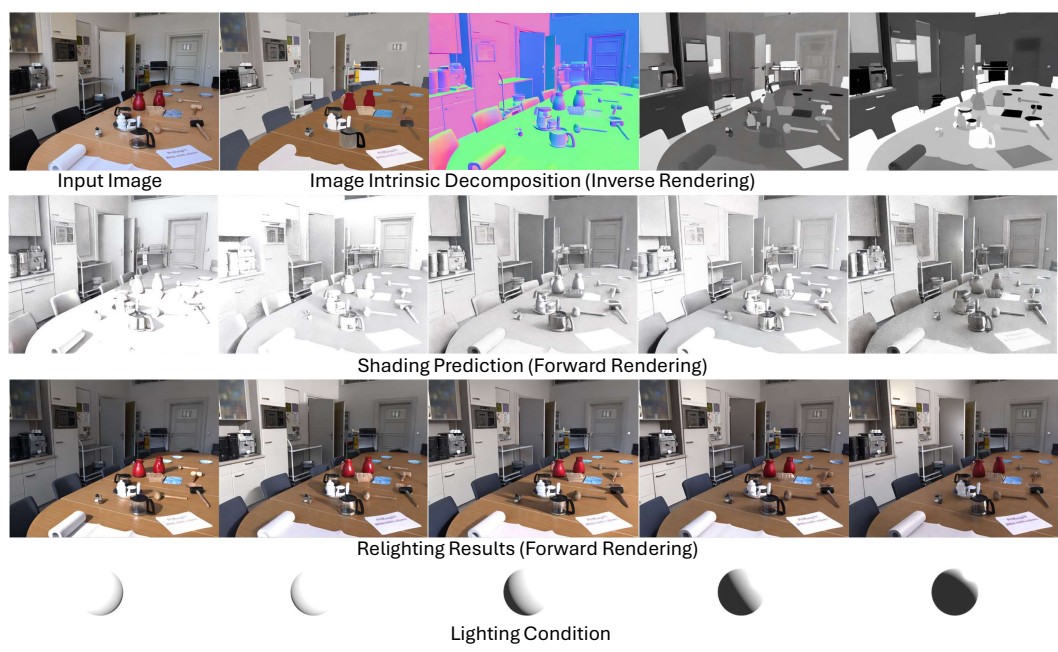

Figure 25: More results showcase the performance of our whole pipeline. Our method successfully relights the glass and accurately predicts the specular highlights.

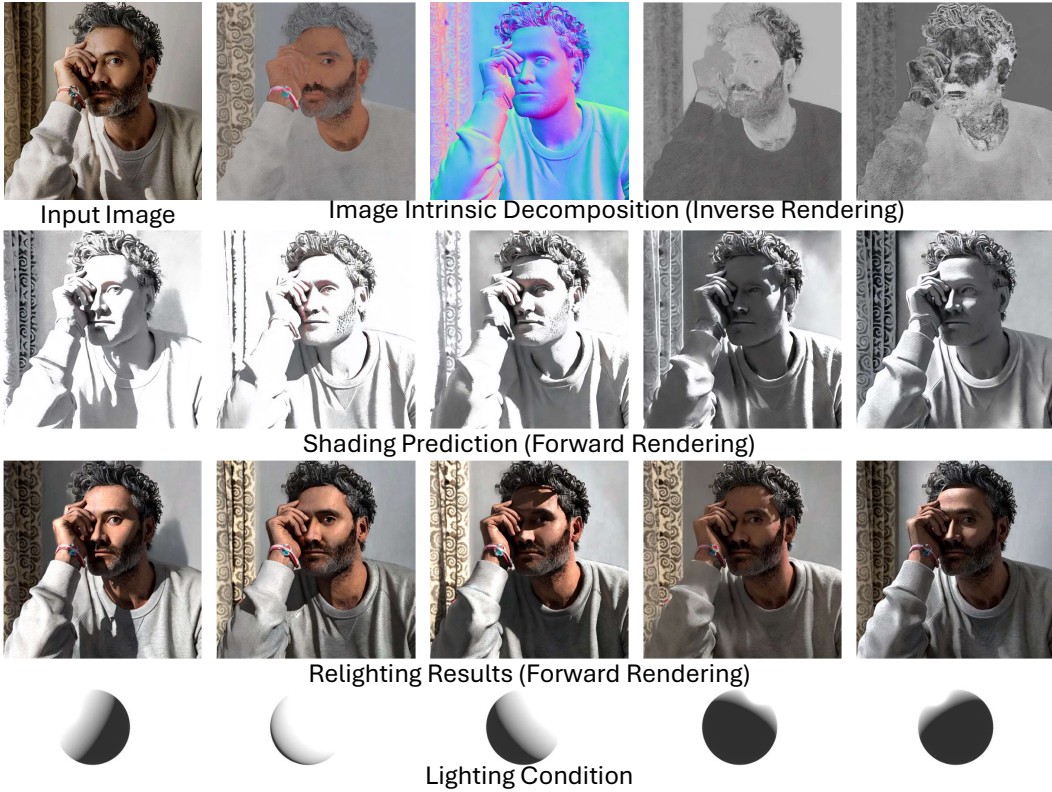

Figure 26: More results showcase the performance of our whole pipeline. Our method can also generalize to human relighting.

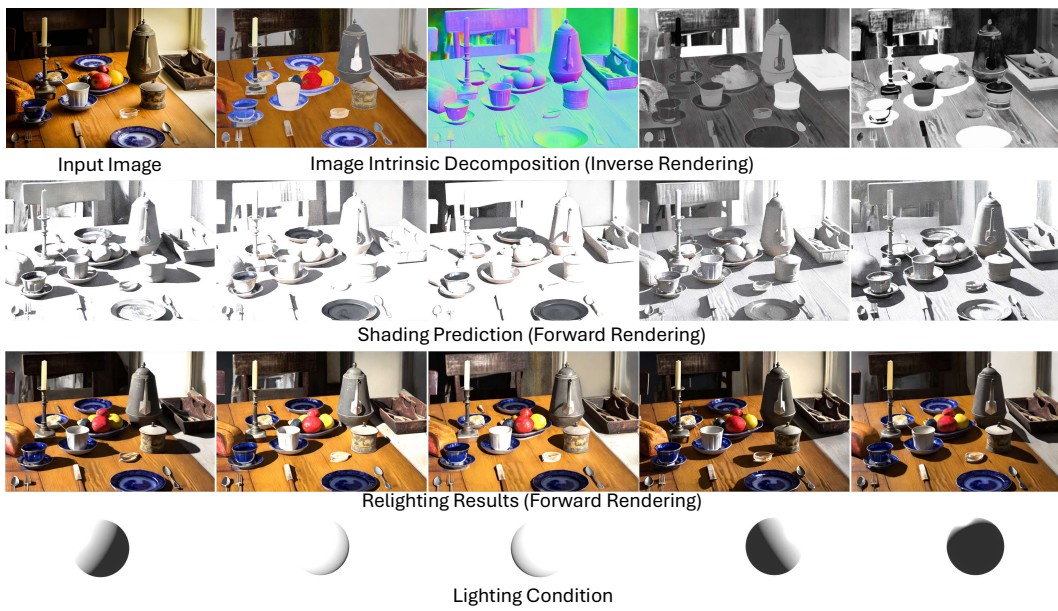

Figure 27: More results showcase the performance of our whole pipeline. Our method successfully generates shadow projections on the tabletop.

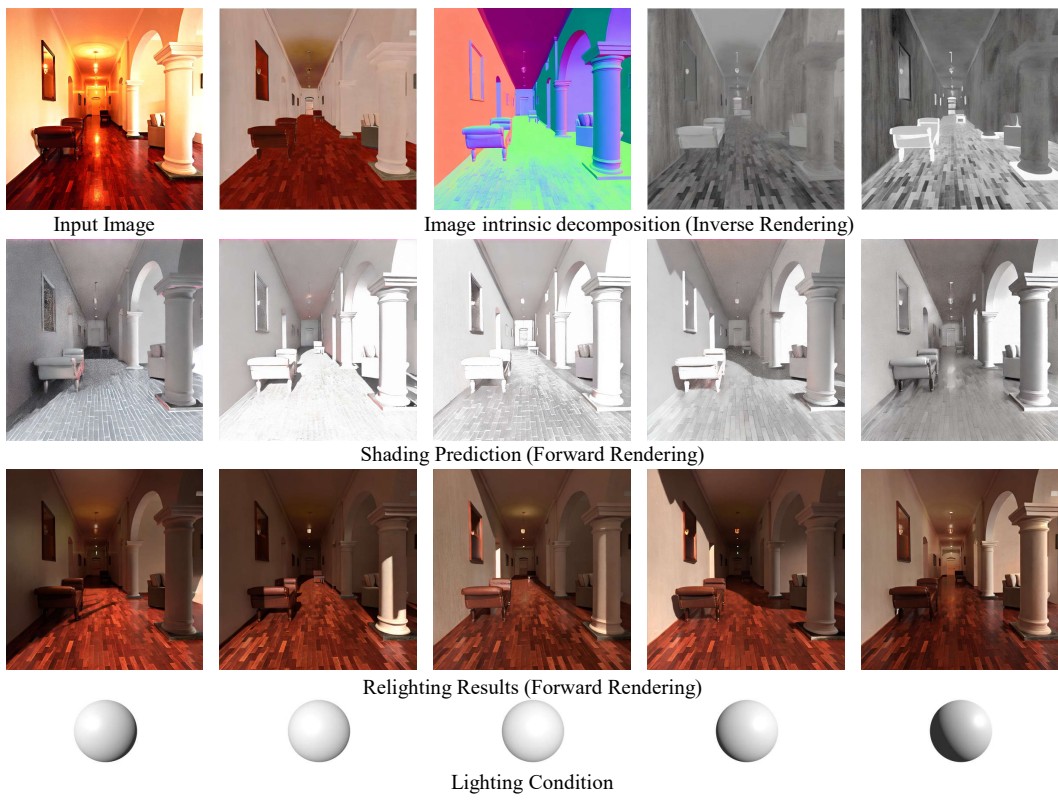

Figure 28: More results showcase the performance of our full pipeline. Our method also generalizes well to indoor scenes with strong built-in light sources.

