# OpenReview forum: "PI-Light: Physics-Inspired Diffusion for Full-Image Relighting"
_ICLR.cc/2026/Conference — ICLR 2026 Poster_

### Official Review · Reviewer_Wtey · 2025-10-28

**Soundness:** 2
**Presentation:** 3
**Contribution:** 2
**Rating:** 6
**Confidence:** 4

**Summary:**

This paper proposes a physics-inspired diffusion for full-image relighting that integrates light transport priors into neural forward rendering to achieve more precise relighting. Additionally, the authors introduce a new dataset comprising diverse objects and scenes.

**Strengths:**

+ Experimental performance: The proposed method demonstrates superior results compared to existing methods in terms of consistency, maintaining, and rendering quality.
+ Clarity and readability: The manuscript is well-structured and clearly written, making the methodology and contributions easy to follow.
+ Attribution: This paper provides a physical perspective to solve the relighting problem, and a meaningful dataset is collected.

**Weaknesses:**

- The authors claim that existing methods struggle to generalize to out-of-distribution data (Line 51). It remains unclear how the proposed approach addresses this challenge. The authors are encouraged to elaborate on the mechanisms or design choices that contribute to improved generalization beyond the training distribution.
- As shown in Fig. 2, the proposed method appears to misinterpret the light reflections of “moonlight scattered on the sea surface” in the background painting and treats it as the light source. The authors may improve albedo estimation accuracy by incorporating a deeper understanding of scene semantics under such challenging conditions.
- The paper should provide more details regarding the trainable components of the proposed network. In particular, it remains unclear which layers (e.g., convolutional layers in Fig. 3) are trainable and what their specific architectures or configurations are.
- Comparison with $RGB \leftrightarrow X$: Since the $RGB \leftrightarrow X$ was trained only on indoor scenes, it results in limited generalization performance. A performance comparison with a version of $RGB \leftrightarrow X$ trained on the same datasets as the proposed method is needed. This would offer a fairer evaluation of the proposed approach’s advantages.

**Questions:**

- About the batch-aware attention: Lines 237-238 state that “the standard self-attention layers are extended to be global-aware, enabling communication across batches. This design improves both efficiency and consistency among predicted intrinsics.” Can you provide a more intuitive explanation of why and how the self-attention can lead to such improvements in consistency and efficiency? And why can the observed performance gains be attributed to self-attention layers? Please provide more details on the implementation of batch-aware attention and how information is exchanged across batches.
- The paper mentions that normal and roughness are fed into the frozen VAE encoder pretrained on RGB images to extract latent features. Given that the encoder was trained on RGB data, have the authors considered the potential domain gap between the intrinsic maps (e.g., normal and roughness) and the RGB domain? How can you ensure the extracted features are practical?

---

> ### Author Response · Authors · 2025-11-21
> **Response to Reviewer Wtey (Part 1)**
>
> Dear Reviewer Wtey, we sincerely thank you for your insightful review. We appreciate that you found our results superior, the manuscript easy to follow, and the physical perspective and dataset meaningful. Regarding the concerns, please find our response below.
>
> ### Regarding weaknesses
>
> - **"The authors claim that existing methods struggle to generalize to out-of-distribution data (Line 51). It remains unclear how the proposed approach addresses this challenge. The authors are encouraged to elaborate on the mechanisms or design choices that contribute to improved generalization beyond the training distribution."**
>
>   Firstly, our dataset contains a variety of objects as well as both indoor and outdoor scenes, which helps improve robustness under distribution shifts. More importantly, as described in Lines 083-089, we design a physics-inspired neural forward rendering module, where
>   the physics-based loss serves as an efficient learning mechanism. This physics-inspired loss function regularizes the training dynamics into a physically plausible landscape, enabling easier convergence and allowing the model to learn more correct light transport with less data
>   and computation.
>
>   To further illustrate this effect, we include qualitative comparisons on out-of-distribution inputs in Fig. 14 (Lines 879–890) of the revised paper, with and without the physics-inspired loss functions. As shown in the examples, incorporating the physics-inspired loss produces more realistic and detailed results, demonstrating its contribution to improved generalization.
>
> - **"As shown in Fig. 2, the proposed method appears to misinterpret the light reflections of “moonlight scattered on the sea surface” in the background painting and treats it as the light source. The authors may improve albedo estimation accuracy by incorporating a deeper understanding of scene semantics under such challenging conditions."**
>
>   We appreciate the reviewer’s insightful observation. The example in Fig. 2 is indeed a challenging case. The background painting contains “moonlight scattered on the sea surface” whose strong highlights visually resemble actual illumination. Since our current intrinsic decomposition model does not incorporate semantic understanding, it may occasionally interpret such stylized reflections as real light sources.
>
>   We agree that integrating richer scene semantics could help the model better distinguish between physically plausible illumination and artistic highlights. Addressing these rare but difficult cases is an interesting direction for future work, and we thank the reviewer for the valuable suggestion.
>
> - **"The paper should provide more details regarding the trainable components of the proposed network. In particular, it remains unclear which layers (e.g., convolutional layers in Fig. 3) are trainable and what their specific architectures or configurations are."**
>
>   All network modules marked with the flame icon in the figures are trainable. The convolution layer shown in Fig. 3 is an additional single convolution layer used to match the channel dimensions between the channel-wise aggregated input and the original U-Net input format in the diffusion model. In summary, the trainable components include: (1) the U-Net in Stage 1 (marked with the flame icon), and (2) in Stage 2, the three convolutional layers and the U-Net, all of which are also marked as trainable in the figure. We have added an icon explanation in the revised Fig.3 for clarity.
>
> - **"Comparison with RGB$\leftrightarrow$X: Since the RGB$\leftrightarrow$X was trained only on indoor scenes, it results in limited generalization performance. A performance comparison with a version of RGB$\leftrightarrow$X trained on the same datasets as the proposed method is needed. This would offer a fairer evaluation of the proposed approach’s advantages."**
>
>   Thank you for the reviewer’s insightful suggestion. Since the official training code of RGB$\leftrightarrow$X is not publicly available, we have tried our best to reproduce the method following the training details in their paper and retrained RGB$\leftrightarrow$X on the same dataset used by our method.
>
>   The evaluation results are reported below (Table 1 in Response Part 2). Even after retraining RGB$\leftrightarrow$X on our dataset, our method consistently achieves better performance, demonstrating that the methodological design itself also contributes to the performance gain.

---

> > ### Author Response · Authors · 2025-11-21
> > **Response to Reviewer Wtey (Part 2)**
> >
> > ### Table 1. Quantitative comparisons with the retrained RGB$\leftrightarrow$X model
> >
> >   | Dataset  | Method            | Albedo PSNR↑ | Albedo SSIM↑ | Albedo LPIPS↓ | Normal PSNR↑ | Normal SSIM↑ | Normal LPIPS↓ | Rough PSNR↑ | Rough SSIM↑ | Rough LPIPS↓ | Metal PSNR↑ | Metal SSIM↑ | Metal LPIPS↓ | Avg PSNR↑ | Avg SSIM↑  | Avg LPIPS↓ |
> >   | -------- | ----------------- | ------------ | ------------ | ------------- | ------------ | ------------ | ------------- | ----------- | ----------- | ------------ | ----------- | ----------- | ------------ | --------- | ---------- | ---------- |
> >   |          | RGB↔X             | 20.09        | 0.9128       | 0.0737        | 22.78        | 0.9039       | 0.0793        | 18.43       | **0.9061**  | 0.0752       | **15.97**   | **0.8404**  | 0.1226       | 19.32     | 0.8908     | 0.0877     |
> >   | Object50 | RGB↔X (retrained) | 19.51        | 0.9071       | 0.0726        | 22.34        | 0.8908       | 0.0846        | 20.47       | 0.8876      | 0.0800       | 15.19       | 0.8309      | 0.1227       | 19.38     | 0.8791     | 0.0900     |
> >   |          | **Ours**          | **22.09**    | **0.9200**   | **0.0562**    | **24.97**    | **0.9217**   | **0.0531**    | **21.09**   | 0.8961      | **0.0736**   | 13.96       | 0.8357      | **0.1216**   | **20.53** | **0.8934** | **0.0761** |
> >   |          | RGB↔X             | 14.07        | 0.6955       | 0.2441        | 14.59        | 0.6824       | 0.3278        | 11.14       | 0.6187      | 0.3086       | **7.20**    | **0.2217**  | 0.6764       | 11.75     | 0.5546     | 0.3892     |
> >   | Scene200 | RGB↔X (retrained) | 13.16        | 0.6659       | 0.2802        | 14.68        | 0.6025       | 0.3876        | 12.08       | 0.5453      | 0.3977       | 6.86        | 0.1928      | **0.6748**   | 11.70     | 0.5016     | 0.4351     |
> >   |          | **Ours**          | **14.46**    | **0.7025**   | **0.2248**    | **16.26**    | **0.7208**   | **0.2768**    | **14.05**   | **0.6348**  | **0.2980**   | 7.02        | 0.2001      | 0.6966       | **12.95** | **0.5646** | **0.3741** |

---

> > > ### Author Response · Authors · 2025-11-21
> > > **Response to Reviewer Wtey (Part 3)**
> > >
> > > ### Regarding questions
> > >
> > > - **"About the batch-aware attention: Lines 237-238 state that “the standard self-attention layers are extended to be global-aware, enabling communication across batches. This design improves both efficiency and consistency among predicted intrinsics.” Can you provide a more intuitive explanation of why and how the self-attention can lead to such improvements in consistency and efficiency? And why can the observed performance gains be attributed to self-attention layers? Please provide more details on the implementation of batch-aware attention and how information is exchanged across batches."**
> > >
> > >   Thanks for the reviewer's valuable suggestion. As mentioned in the paper, we follow the design used in Wonder3D (Long et al., 2024) and GeoWizard(Fu et al., 2024) for the cross-batch self-attention layer, and therefore did not provide a detailed explanation in the main text. For clarity, we describe the computation here: in the cross-batch self-attention layers, the queries, keys, and values are computed as follows:
> > >   $$
> > >   q_a=W_q \hat{z}^a,k_a=W_k(\hat{z}^a\oplus\hat{z}^n\oplus\hat{z}^r\oplus\hat{z}^m),v_a=W_v(\hat{z}^a\oplus\hat{z}^n\oplus\hat{z}^r\oplus\hat{z}^m)
> > >   $$
> > >
> > >   $$
> > >   q_n=W_q \hat{z}^n,k_n=W_k(\hat{z}^n\oplus\hat{z}^a\oplus\hat{z}^r\oplus\hat{z}^m),v_n=W_v(\hat{z}^n\oplus\hat{z}^a\oplus\hat{z}^r\oplus\hat{z}^m)
> > >   $$
> > >
> > >   $$
> > >   q_r=W_q\hat{z}^r,k_r=W_k(\hat{z}^r\oplus\hat{z}^a\oplus\hat{z}^n\oplus\hat{z}^m),v_r=W_v(\hat{z}^r\oplus\hat{z}^a\oplus\hat{z}^n\oplus\hat{z}^m)
> > >   $$
> > >
> > >   $$
> > >   q_m=W_q \hat{z}^m,k_m=W_k(\hat{z}^m\oplus\hat{z}^a\oplus\hat{z}^r\oplus\hat{z}^n),v_m=W_v(\hat{z}^m\oplus\hat{z}^a\oplus\hat{z}^r\oplus\hat{z}^n)
> > >   $$
> > >
> > >   where, $\hat{z}^a,\hat{z}^n,\hat{z}^r,\hat{z}^m$ are latent albedo, normal, roughness and metallic embeddings in transformer blocks. The operator $\oplus$ denotes concatenation, and $W_q$, $W_k$ and $W_v$ are the projection matrices used to compute the query, key, and value embeddings. The cross-batch features are $Att(q_i,k_i,v_i),i = \{a,n,r,m\}$, where $Att()$ denotes the standard softmax attention.  This design enforces stronger consistency among albedo, normal, roughness, and metallic predictions. In terms of efficiency, compared to the prior method RGB$\leftrightarrow$X, which produces one intrinsic map at a time, our framework predicts all intrinsic components simultaneously. As a result, our inference time is lower: RGB$\leftrightarrow$X requires approximately 9 seconds per image, whereas our method takes about 4 seconds per image on a single H200 GPU. We appreciate the reviewer’s suggestion. We have added this explanation, along with an illustrative figure (Fig.13 in Lines 843-855) to the revised paper to further clarify the cross-batch self-attention layer.
> > >
> > > - **"The paper mentions that normal and roughness are fed into the frozen VAE encoder pretrained on RGB images to extract latent features. Given that the encoder was trained on RGB data, have the authors considered the potential domain gap between the intrinsic maps (e.g., normal and roughness) and the RGB domain? How can you ensure the extracted features are practical?"**
> > >
> > >   Normal and albedo are multi-channel inputs, and for single-channel maps such as roughness and metallic, we replicate the channel to form a three-channel input. In this way, the frozen VAE encoder treats them similarly to grayscale images. Since the VAE was trained on a diverse RGB dataset that naturally includes grayscale images, this mapping does not cause a significant domain mismatch.
> > >
> > >   To further verify that the extracted features are practical, we compute the reconstruction quality by passing all intrinsic maps from our Obj50 test set through the frozen VAE encoder and then decoding them back. As shown in the table below, the reconstructed results achieve PSNR values higher than 30 dB, and visually the differences before and after VAE reconstruction are barely noticeable. This demonstrates that the frozen VAE can effectively encode these intrinsic maps without introducing substantial degradation.
> > >
> > >   | Intrinsic type | Raw Image | Albedo | Normal | Metallic | Roughness |
> > >   | -------------- | --------- | ------ | ------ | -------- | --------- |
> > >   | **PSNR**       | 36.41     | 38.57  | 33.80  | 39.21    | 41.86     |

---

> > > > ### Comment · Reviewer_Wtey · 2025-11-26
> > > >
> > > > Thanks for the authors’ responses. They address part of my concerns regarding the comparison with RGB→X and demonstrate improved performance. For the challenging case shown in Fig. 2, the authors are encouraged to incorporate semantic understanding into their modeling. Therefore, I will keep my score at 6.

---

### Official Review · Reviewer_GfkN · 2025-10-31

**Soundness:** 3
**Presentation:** 2
**Contribution:** 3
**Rating:** 8
**Confidence:** 5

**Summary:**

This paper tackles the problem of full-image relighting. To enable realistic relighting results, the authors propose to first estimate intrinsic properties, e.g., albedo, normal, and roughness, based on a diffusion model. Later, these intrinsic properties are fed into another diffusion model that produces a relit image as well as diffuse and specular maps. The inputs for these outputs are inspired by physically-based rendering. Specifically, one takes the source image and albedo; one takes normal, lighting, and mask; the other uses normal, lighting, metallic, roughness, and mask. Further, with a simplified physically based rendering equation, i.e., albedo $\cdot$ diffuse + specular, the directly-produced relit image is compared against the physically-rendering-composed image. Experiments on various datasets verify the effectiveness.

**Strengths:**

- originality-wise: the idea of closely following physically-based rendering to construct the input for the diffusion model is interesting.
- quality-wise: the qualitative and quantitative results are promising.
- clarity-wise: the presentation is good in general but can be further improved.
- significance-wise: the realistic relighting is important for various downstream applications, e.g., AR/VR.

**Weaknesses:**

## 1. Clarification on image intrinsics estimation

Can authors clarify why the model in the paper works better than baselines, e.g., IntrinsicAnything in Tab.1? Is it because of carefully curated data or different model designs?

## 2. About efficiency

**For training**. As mentioned in L302:
> Since physical laws lose their meaning when computed in the latent space, our physics-based losses for the relighting model are applied entirely in the RGB space, i.e., it is computed after the VAE decoder reconstructs the output

This makes the training extremely costly as each data in the training set needs to go through the full diffusion process to obtain the final latent. How to make the training affordable?

**For inference**, can the author provide some runtime analysis for the inference?

## 3. Presentation

1. There is no lighting condition in Figure 3's "Stage 2"

2. For Eq. (6) and (8), the input and output orders are not aligned. Either the authors can have separate equations, or the authors can make the order aligned. Currently, it isn't very clear. I do not know which input corresponds to each output.

## 4. References

Missing related works for generative relighting

[a] Ginter et al., A Diffusion Approach to Radiance Field Relighting using Multi-Illumination Synthesis. EGSR 2024.

[b] Zhao et al., IllumiNeRF 3D Relighting without Inverse Rendering. NeurIPS 2024.

**Questions:**

See "Weakness"

---

> ### Author Response · Authors · 2025-11-21
> **Response to Reviewer GfkN (Part 1)**
>
> Dear Reviewer GfkN,  we sincerely thank you for the insightful and valuable comments. We appreciate that you found our idea interesting, our results promising and the realistic relighting is important for various downstream tasks. Regarding the concerns, please find our response below.
>
> 1. **"Can authors clarify why the model in the paper works better than baselines, e.g., IntrinsicAnything in Tab.1? Is it because of carefully curated data or different model designs?"**
>
>     Both the proposed dataset and our model designs contribute to the performance improvements. In addition to the ablation studies provided in Table 3 of our paper, we also retrained a state-of-the-art method, RGB$\leftrightarrow$X, on the same dataset used in our work. The results, shown in the table below (Table 1 in Response Part 2), indicate that our model design consistently outperforms RGB$\leftrightarrow$X under identical training data conditions, demonstrating that the gains are not solely due to curated data but also stem from our architectural and loss-design choices.
>
> 2. **"This makes the training extremely costly as each data in the training set needs to go through the full diffusion process to obtain the final latent. How to make the training affordable?"**
>
>     Using Eq. 3 in Lines 174–178 in our paper, we can derive an equation $z_0=\sqrt{\bar{\alpha}_t}z_t-\sqrt{1-\bar{\alpha}_t}v_t$ that allows us to recover $z_0$ directly from the current latent $z_t$ and the predicted $v_t$. This means that the full diffusion process does not need to be executed for training. For every training sample, the latent $z_0$ can be obtained in a single step at the chosen timestep $t$, and then passed to the VAE decoder.
>
> 3. **"Training and inference efficiency"**
>
>     The training of each stage takes approximately 20 hours on four H200 GPUs. During inference, Stage 1 requires about 4 seconds per image and Stage 2 requires about 5 seconds per image on a single H200 GPU. For comparison, one of the state-of-the-art intrinsic decomposition methods, RGB$\leftrightarrow$X, takes around 9 seconds per image for each stage on a single H200 GPU.
>
> 4. **"There is no lighting condition in Figure 3's Stage 2"**
>
>     The gray ball in Figure 3 represents the lighting condition. We acknowledge that this may not have been clearly indicated in the original submission. We have updated Figure 3 to make the lighting condition more explicit.
>
> 5. **"For Eq. (6) and (8), the input and output orders are not aligned. Either the authors can have separate equations, or the authors can make the order aligned. Currently, it isn't very clear. I do not know which input corresponds to each output."**
>
>     $I_{in1},I_{in2},I_{in3}$ in Eq. (6) correspond to the outputs $I_{relit}$, $D_{pred}$, $S_{pred}$ in Eq. (8), respectively. Specifically, $I_{in1}=(I_{in},A)\rightarrow I_{relit}$, $I_{in2}=(N,L,m)\rightarrow D_{pred}$, $I_{in3}=(N,L,M,R,m)\rightarrow S_{pred}$. These three input batches are also illustrated in Stage 2 of Fig. 3. Following the reviewer’s suggestion, we have added an additional illustration in this section to make this correspondence clearer.
>
> 6. **"Reference"**
>
>     We thank the reviewer for the suggestion. The two missing references have now been added to the main paper.

---

> ### Author Response · Authors · 2025-11-21
> **Response to Reviewer GfkN (Part 2)**
>
> ### Table 1. Quantitative comparisons with the retrained RGB$\leftrightarrow$X model (for Weakness 1).
>  | Dataset  | Method            | Albedo PSNR↑ | Albedo SSIM↑ | Albedo LPIPS↓ | Normal PSNR↑ | Normal SSIM↑ | Normal LPIPS↓ | Rough PSNR↑ | Rough SSIM↑ | Rough LPIPS↓ | Metal PSNR↑ | Metal SSIM↑ | Metal LPIPS↓ | Avg PSNR↑ | Avg SSIM↑  | Avg LPIPS↓ |
>   | -------- | ----------------- | ------------ | ------------ | ------------- | ------------ | ------------ | ------------- | ----------- | ----------- | ------------ | ----------- | ----------- | ------------ | --------- | ---------- | ---------- |
>   |          | RGB↔X             | 20.09        | 0.9128       | 0.0737        | 22.78        | 0.9039       | 0.0793        | 18.43       | **0.9061**  | 0.0752       | **15.97**   | **0.8404**  | 0.1226       | 19.32     | 0.8908     | 0.0877     |
>   | Object50 | RGB↔X (retrained) | 19.51        | 0.9071       | 0.0726        | 22.34        | 0.8908       | 0.0846        | 20.47       | 0.8876      | 0.0800       | 15.19       | 0.8309      | 0.1227       | 19.38     | 0.8791     | 0.0900     |
>   |          | **Ours**          | **22.09**    | **0.9200**   | **0.0562**    | **24.97**    | **0.9217**   | **0.0531**    | **21.09**   | 0.8961      | **0.0736**   | 13.96       | 0.8357      | **0.1216**   | **20.53** | **0.8934** | **0.0761** |
>   |          | RGB↔X             | 14.07        | 0.6955       | 0.2441        | 14.59        | 0.6824       | 0.3278        | 11.14       | 0.6187      | 0.3086       | **7.20**    | **0.2217**  | 0.6764       | 11.75     | 0.5546     | 0.3892     |
>   | Scene200 | RGB↔X (retrained) | 13.16        | 0.6659       | 0.2802        | 14.68        | 0.6025       | 0.3876        | 12.08       | 0.5453      | 0.3977       | 6.86        | 0.1928      | **0.6748**   | 11.70     | 0.5016     | 0.4351     |
>   |          | **Ours**          | **14.46**    | **0.7025**   | **0.2248**    | **16.26**    | **0.7208**   | **0.2768**    | **14.05**   | **0.6348**  | **0.2980**   | 7.02        | 0.2001      | 0.6966       | **12.95** | **0.5646** | **0.3741** |

---

### Official Review · Reviewer_wQ6F · 2025-11-01

**Soundness:** 3
**Presentation:** 3
**Contribution:** 3
**Rating:** 6
**Confidence:** 2

**Summary:**

This paper proposes PI-Light, a framework to relight images via the extraction of intrinsic properties of a scene and diffusion-based rendering for the final result: in the inverse rendering stage, surface normal and materials are estimated from a customized diffusion model; in the neural forward rendering stage, another diffusion model is used to synthesize the image under target lighting conditions. To facilitate training of the models, the authors constructed a new dataset featuring rendered data using objects from Objaverse. In the experiment section, the authors compare PI-Light with both intrinsic decomposition models and relighting methods. The proposed method shows reasonable performance.

**Strengths:**

- The motivation for having physics-inspired scene properties in the relighting pipeline is solid. With more and more frameworks becoming end-to-end, it is good to have a variety of approaches with competitive performance. In particular, this paper shows that it is possible to combine intrinsic decomposition with diffusion-based relighting for a good result.

- The paper focuses on full image relighting, which is an under-explored direction in image relighting, although currently both the dataset and the experiment in this paper are still mainly object-centric.

- The paper is easy to follow, and the design of the framework is well described.

- The proposed dataset could be valuable to researchers studying both image relighting and intrinsic decomposition tasks.

**Weaknesses:**

- Intrinsic decomposition using neural networks is a well-studied problem. There is little new insight from this paper (Section 4.2).

- It seems the relighting results shown in the paper are mostly directional lighting. In other relighting work, such as Neural Gaffer, HDR image-based environment maps are used as lighting conditions which provide more natural and complex lighting conditions. Since the proposed method is targeting full image relighting, single directional lighting could be a limitation.

- Evaluation is done on 50 objects and 20 scenes. It might be better to try with more objects and scenes, as well as on other relighting datasets.

**Questions:**

- Why is the full model performance in Table 3 (last row) different from Table 2 (last row)? If I understand it correctly, they should be the same model and tested on the same Object50 test set.

- How does the proposed method work with HDR image-based environment maps? It seems by design the pipeline should not be limited to directional lights.

---

> ### Author Response · Authors · 2025-11-21
> **Response to Reviewer wQ6F (Part 1)**
>
> Dear Reviewer wQ6F, we sincerely thank you for your insightful comments. We appreciate you mentioning that our motivation is solid, the framework has variety, the performance is competitive and the dataset could be valuable to the community. Regarding the concerns, please find our response below.
>
> ### Regarding weaknesses
>
> 1. **"Intrinsic decomposition using neural networks is a well-studied problem. There is little new insight from this paper (Section 4.2)."**
>
>     Thanks for the reviewer’s comments. We acknowledge that intrinsic decomposition using neural networks has indeed been explored by many prior works. However, we would like to clarify that our pipeline still differs in several meaningful aspects. As stated in Lines 081–083 of the Introduction, unlike prior state-of-the-art intrinsic decomposition approaches, we extend standard self-attention layers to be global-aware. Moreover, intrinsic decomposition is not the main contribution of our work. As highlighted in Lines 099–107, our key innovations lie in the newly constructed dataset, the neural forward-rendering pipeline, and the physics-inspired loss functions. Together, these components differentiate our framework from related prior approaches.
>
> 2. **"It seems the relighting results shown in the paper are mostly directional lighting. In other relighting work, such as Neural Gaffer, HDR image-based environment maps are used as lighting conditions which provide more natural and complex lighting conditions. Since the proposed method is targeting full image relighting, single directional lighting could be a limitation."**
>
>     Under the principle of linear light superposition, an image rendered under a single directional light can be treated as one OLAT (One Light At a Time) basis image. By rendering multiple basis images from different directions and applying appropriate weights, we can synthesize the image under any HDR environment map, similar to the OLAT procedure used in prior works.
>
> 3. **"Evaluation is done on 50 objects and 20 scenes. It might be better to try with more objects and scenes, as well as on other relighting datasets."**
>
>     Each object in our test set is rendered from 4 viewpoints under 6 different lighting conditions, resulting in 1200 images even for the original set of 50 objects. Following the reviewer's suggestion, we have expanded our test split from 50 objects / 20 scenes to 500 objects / 36 scenes, that is, obj500 contains 500 objects resulting in 12000 images, and the updated quantitative results are reported below (Table 1 in Response Part 2).
>
>     Regarding the use of other relighting datasets, we would like to clarify that, as discussed in Lines 210–214, the commonly used renderer tool, BlenderProc, exhibits light-dependent fluctuations in the predicted albedo. In addition, for transparent or semi-transparent materials, Blender typically outputs saturated black/white albedo rather than the correct material albedo. Existing relighting datasets inherit this issue because they lack corresponding ground-truth labels for these materials, which is one of the motivations behind constructing our new dataset.
>
>     Nevertheless, we are very open to evaluating on additional datasets. If there are specific datasets the reviewer would like us to include, please let us know during the rebuttal period and we will do our best to run the corresponding experiments.
>
> ### Regarding questions
>
> 1. **"Why is the full model performance in Table 3 (last row) different from Table 2 (last row)? If I understand it correctly, they should be the same model and tested on the same Object50 test set."**
>
>     As noted in the caption of Table 3 in our paper, all ablation experiments were trained for 30k iterations, whereas the full model in Table 2 in our paper was trained for 90k iterations, as stated in Lines 373–375. This difference was due to limited GPU availability at the time of preparing the ablation table. We originally believed that 30k iterations were sufficient to reflect the relative contribution of each component.
>
>     To address the reviewer’s concern, we have now retrained all ablation variants for 90k iterations, matching the full model’s training schedule. The updated ablation results are provided below (Table 2 in Response Part 2), and they consistently show that each proposed component contributes to the overall performance gain.
>
> 2. **"How does the proposed method work with HDR image-based environment maps? It seems by design the pipeline should not be limited to directional lights."**
>
>     Theoretically, our pipeline supports HDR environment maps. We currently demonstrate directional lights because most of our data are rendered under directional illumination. Using OLAT, we can render multiple basis images under different directional lights and then combine them with appropriate weights to synthesize renderings under any HDR environment map.

---

> > ### Author Response · Authors · 2025-11-21
> > **Response to Reviewer wQ6F (Part 2)**
> >
> > ### Table 1. Quantitative results on the extended dataset (for Weakness 3).
> >   | Dataset   | Method            | Albedo PSNR↑ | Albedo SSIM↑ | Albedo LPIPS↓ | Normal PSNR↑ | Normal SSIM↑ | Normal LPIPS↓ | Rough PSNR↑ | Rough SSIM↑ | Rough LPIPS↓ | Metal PSNR↑ | Metal SSIM↑ | Metal LPIPS↓ | Avg PSNR↑ | Avg SSIM↑  | Avg LPIPS↓ |
> >   | --------- | ----------------- | ------------ | ------------ | ------------- | ------------ | ------------ | ------------- | ----------- | ----------- | ------------ | ----------- | ------------ | ------------ | --------- | ---------- | ---------- |
> >   |           | IntrinsicAnything | 15.78        | 0.8535       | 0.1191        | -            | -            | -             | -           | -           | -            | -           | -            | -            | -         | -          | -          |
> >   |           | Kocsis et al.*    | 19.85        | 0.8997       | 0.0852        | **24.93**    | **0.9350**   | 0.0665        | 16.75       | 0.8877      | 0.0804       | 14.14       | **0.8435**   | 0.1245       | 18.92     | **0.8915** | 0.0892     |
> >   | Object500 | Zhu et al.*       | 19.17        | 0.8681       | 0.0933        | 21.60        | 0.8817       | 0.0903        | 15.98       | 0.8500      | 0.1082       | 12.23       | 0.8140       | 0.1572       | 17.25     | 0.8535     | 0.1123     |
> >   |           | RGB↔X             | 19.86        | 0.9015       | 0.0807        | 22.78        | 0.9089       | 0.0862        | 17.27       | **0.8881**  | 0.0773       | **16.20**   | 0.8408       | 0.1261       | 19.03     | 0.8848     | 0.0926     |
> >   |           | **Ours**          | **20.47**    | **0.9039**   | **0.0661**    | 24.61        | 0.9261       | **0.0594**    | **19.51**   | 0.8859      | **0.0793**   | 14.90       | 0.8384       | **0.1199**   | **19.87** | 0.8886     | **0.0812** |
> >
> >   | Dataset | Method            | Albedo PSNR↑ | Albedo SSIM↑ | Albedo LPIPS↓ | Normal PSNR↑ | Normal SSIM↑ | Normal LPIPS↓ | Rough PSNR↑ | Rough SSIM↑ | Rough LPIPS↓ | Metal PSNR↑ | Metal SSIM↑ | Metal LPIPS↓ | Avg PSNR↑ | Avg SSIM↑  | Avg LPIPS↓ |
> >   | ------- | ----------------- | ------------ | ------------ | ------------- | ------------ | ------------ | ------------- | ----------- | ----------- | ------------ | ----------- | ------------ | ------------ | --------- | ---------- | ---------- |
> >   |         | IntrinsicAnything | 13.10        | 0.6175       | 0.3655        | -            | -            | -             | -           | -           | -            | -           | -            | -            | -         | -          | -          |
> >   |         | Kocsis et al.*    | **14.49**    | 0.6793       | 0.2854        | **18.37**    | **0.7578**   | **0.2634**    | 9.92        | 0.5988      | 0.3269       | 5.34        | **0.2309**   | **0.5901**   | 12.03     | **0.5667** | 0.3665     |
> >   | Scene36 | Zhu et al.*       | 13.61        | 0.6408       | 0.2630        | 16.32        | 0.6694       | 0.2648        | 9.30        | 0.5262      | 0.3301       | 5.08        | 0.2076       | 0.6037       | 11.08     | 0.5110     | **0.3654** |
> >   |         | RGB↔X             | 13.81        | 0.6807       | 0.2629        | 14.64        | 0.6675       | 0.3424        | 10.92       | 0.6078      | 0.3190       | **7.11**    | 0.2151       | 0.6749       | 11.62     | 0.5428     | 0.3998     |
> >   |         | **Ours**          | 14.10        | **0.6852**   | **0.2428**    | 16.24        | 0.7122       | 0.2757        | **13.60**   | **0.6151**  | **0.3129**   | 6.97        | 0.2021       | 0.6915       | **12.73** | 0.5537     | 0.3807     |
> >
> >   \* indicates resolution differs from original 768×768 test dataset.  Kocsis et al. resize to 480×640; Zhu et al. resize to 240×320.
> >
> > ****
> >
> >   ### Table 2. Updated ablation studies (for Question 1).
> >
> >   | Decomposition | $L_{DS}$ | $L_{PS}$ | PSNR ↑    | SSIM ↑     | LPIPS ↓    |
> >   | ------------- | -------- | -------- | --------- | ---------- | ---------- |
> >   |               |          |          | 11.64     | 0.9055     | 0.0629     |
> >   | ✓             |          |          | 12.31     | 0.9118     | 0.0519     |
> >   | ✓             | ✓        |          | 12.89     | 0.9124     | 0.0507     |
> >   | ✓             |          | ✓        | 13.42     | 0.9152     | **0.0456**     |
> >   | ✓             | ✓        | ✓        | **14.09** | **0.9211** | 0.0553 |

---

### Author Response · Authors · 2025-12-03
**Author Final Remarks**

We sincerely thank all reviewers for their time and thoughtful feedback and thank AC for the dedication to the review process. We are encouraged by the uniformly positive recommendations from the reviewers.

**Strengths.** Reviewers highlighted that our motivation is solid (**wQ6F**), the framework is diverse and shows the possibility to combine intrinsic decomposition with diffusion-based relighting (**wQ6F**); the task is under-explored (**wQ6F**) and important for various downstream applications (**GfkN**); the dataset could be valuable to the community (**wQ6F**); the idea of closely following physically-based rendering to construct the input for the diffusion model is interesting (**GfkN**); the results are promising (**GfkN**, **Wtey**); the presentation is good, well-structured and clearly written(**GfkN**, **Wtey**); our approach provides a physical perspective to solve the relighting problem (**Wtey**); the dataset is meaningful (**Wtey**).

**Rebuttal.** In the rebuttal, we addressed the major concerns:
1. **Retraining RGB$\leftrightarrow$X on our proposed dataset (GfkN, Wtey)**: We conducted additional experiments to reproduce and retrain the most similar work, RGB$\leftrightarrow$X, on our proposed dataset, and compared it with our model to demonstrate that the performance gain comes not only from the dataset but also from our model design;
2. **How we address generalization to out-of-distribution data (Wtey)**: We included more qualitative results to show that the physical-inspired loss of our pipeline enables strong generalization to out-of-distribution data;
3. **Test set size (wQ6F)**: We expanded the test dataset from 50 objects / 20 scenes to 500 objects / 36 scenes (12,000 images in total) and provided evaluations on this extended test set;
4. **Writing quality (GfkN, Wtey)**: We further clarified details of our methods and revised ambiguous descriptions in the paper.

Given the reviewers’ positive assessments  (6, 8, 6) and our clarifications above, we respectfully hope the committee will recognize the value of our contributions to the research community.

---

### Meta-Review · Area_Chair_ttmN · 2026-01-06

**Summary:**

Following is a summary of the reviewers' major concerns:

### Reviewer wQ6F
1. Little new insight on the long-standing problem of intrinsic decomposition.
    * **Author replies**: Intrinsic decomposition is not the main contribution of the paper.
    The paper focuses more on the new dataset and the new neural rendering pipeline.
2. Most relighting results are directional lights; there are no complex lighting conditions such
as envmap lighting.
    * **Author replies**: Relighting under more complex lighting conditions can be achieved
    via combining the results of multiple directional lights.

### Reviewer GfkN
1. What makes the proposed method outperform baseline methods? Dataset or model designs.
    * **Author replies**: The authors retrained the baseline on the new dataset. The results
    show that both the dataset and model designs help improve the performance.


### Reviewer Wtey
1. How does the method handle OOD data?
    * **Author replies**: Both the new diverse dataset and physics-inspired neural forward rendering module
    help improve OOD performance.
2. The baseline method needs to be retrained on the proposed dataset for fair comparison.
    * **Author replies**: The authors retrained RGB<->X on the proposed dataset and show that the proposed method outperforms
    the retrained baseline.

**Reviewer Concerns:**

See above.

I think the concerns from the reviewers are well addressed.

**Reviewer Scores:**

Reviewer wQ6F: maintain the score

Reviewer GfkN: maintain the score

Reviewer Wtey: maintain the score

---

### Decision · Program_Chairs · 2026-01-26

Accept (Poster)